# Stable iron isotope signals indicate a "pseudo-abiotic" process driving deep iron release in methanic sediments

**Susann Henkel[1], Bo Liu[1], Michael Staubwasser[2], Simone A. Kasemann[3,4], Anette Meixner[3,4], David A. Aromokeye[5,6], Michael W. Friedrich[3,5], and Sabine Kasten[1,3,4]**

[1]Geosciences Department, Alfred Wegener Institute, Helmholtz Centre for Polar and Marine Research, Am Handelshafen 12, 27570 Bremerhaven, Germany

[2]Institute of Geology and Mineralogy, University of Cologne, Zülpicher Str. 49a, 50674 Cologne, Germany

[3]MARUM – Center for Marine Environmental Sciences, University of Bremen, Leobener Str. 8, 28359 Bremen, Germany

[4]Faculty of Geosciences, University of Bremen, Klagenfurter Str., 28359 Bremen, Germany

[5]Faculty of Biology/Chemistry, University of Bremen, James-Watt-Str. 1, 28359 Bremen, Germany

[6]Environment Department, World Bank, Washington, DC 20433, USA

**Correspondence:** Susann Henkel (susann.henkel@awi.de)

**Abstract.** The low $\delta^{56}$Fe values of dissolved iron liberated by microbial iron reduction are characteristic of many shallow subsurface sediments and – if not significantly changed within the oxic sediment layer – the related benthic Fe fluxes into the water column. Here, we decipher whether stable Fe isotope signatures in pore water and the respective solid-phase sediment samples are also useful for unraveling the processes driving Fe liberation in deeper methanic sediments. We investigated the fine-grained deposits of the Helgoland mud area, North Sea, where Fe reduction in the methanic subsurface sediments was previously suggested to be coupled to methanogenic fermentation of organic matter and anaerobic methane oxidation. In the evaluated subsurface sediments, a combination of iron isotope geochemistry with reactive transport modeling for the deeper methanic sediments hints at a combination of processes affecting the stable isotope composition of dissolved iron. However, the dominant process releasing Fe at depth does not seem to lead to notable iron isotope fraction. Under the assumption that iron reducing microbes generally prefer isotopically light iron, the deep Fe reduction in this setting appears to be "pseudo-abiotic": if fermentation is the main reason for Fe release at depth, the fermenting bacteria transfer electrons directly or indirectly to Fe(III), but our data do not indicate notable related isotopic fractionation. Our findings strongly contribute to the debate on the pathway for deep $Fe^{2+}$ release by showing that the main underlying process is mechanistically different to the microbial Fe reduction dominating in the shallow sediments and encourages future studies to focus on the fermentative degradation of organic matter as a source of iron in methanic sediments.

## 1 Introduction

Iron reduction in coastal and marine sediments plays an important role in the degradation of organic matter, the transformation and cycling of carbon species, and benthic nutrient release into the water column (e.g., Baloza et al., 2022; LaRowe and Van Capellen, 2011; Lovley and Phillips, 1986; Thamdrup et al., 1994; Zhou et al., 2023, 2024). Multiple studies, e.g., Henkel et al. (2016, 2018), Johnson and Beard (2005), Severmann et al. (2006), and Staubwasser et al. (2006), have shown that the difference in the isotopic composition of solid Fe(III) and dissolved Fe(II) in shallow marine sediments is similar to the fractionation related to dissimilatory iron reduction (DIR): based on pure-culture studies, dissimilatory iron-reducing microorganisms (e.g., *Shewanella* spp.) favor the light iron isotope $^{54}$Fe, which is therefore preferentially released into pore water while the ferric substrate becomes isotopically heavier (e.g., Beard et al., 1999, 2003a; Johnson et al., 2004, 2005). Part of the micro-

bially liberated (isotopically light) Fe(II) is adsorbed onto the oxide surface and in isotopic exchange with the heavier reactive Fe(III) layer of the oxide. The resulting fractionation (combining DIR and the electron and atom exchange) $\Delta^{56}Fe_{Fe(II)diss-Fe(III)}$ is up to $-3\,‰$ (e.g., Crosby et al., 2005, 2007). Iron isotopes, expressed as $\delta^{56}Fe$ (‰), are thus considered a tool for assessing the role of microbial iron reduction (MIR) for the mineralization of organic matter and for tracing benthic iron fluxes into the water column (e.g., Conway and John, 2014; Homoky et al., 2009; John et al., 2012; Severmann et al., 2006, 2010; Sieber et al., 2021). Here, we aim to evaluate whether pore-water and solid-phase Fe isotope signatures are also useful for unraveling the processes driving Fe reduction in deeper sediments below the sulfate–methane transition (SMT) that is frequently observed in freshwater, brackish, and marine depositional environments (e.g., Egger et al., 2017; Hensen et al., 2003; Kasten et al., 1998; März et al., 2008; Oni et al., 2015a; Riedinger et al., 2005, 2010, 2014; Segarra et al., 2013; Sivan et al., 2011; Wersin et al., 1991). The processes responsible have not been entirely understood so far. Most of the sites at which this "deep Fe reduction" occurs are in high-deposition areas characterized by a rapid transition of Fe-(oxyhydr)oxides through the upper zone of Fe reduction and the following sulfidic interval into methanic, non-sulfidic sediments (e.g., Aromokeye et al., 2020, 2021; Oni et al., 2015a; Riedinger et al., 2005). Dissolved $Fe^{2+}$ concentrations typically increase below the sulfidic interval surrounding the SMT and may reach several hundreds of micromolar, often exceeding Fe concentrations in the upper ferruginous zone close to the sediment surface (e.g., Riedinger et al., 2005, 2014, 2017).

There are a variety of possible biotic and abiotic pathways for deep Fe reduction. Biotic pathways include continuing DIR by use of organic or inorganic electron donors (e.g., Lovley, 1991; Lovley et al., 1989; Roden and Lovley, 1993), organoclastic fermentative Fe reduction (e.g., Lehours et al., 2010; Lovley and Phillips, 1986), Fe reduction coupled to ammonium oxidation (Bao and Li, 2017), and Fe-coupled anaerobic oxidation of methane (Fe-AOM) (e.g., Aromokeye et al., 2020; Beal et al., 2009; Riedinger et al., 2014; Sivan et al., 2011). It was furthermore discussed whether $Fe^{2+}$ release can also be linked to iron oxide reduction by methanogens that can perform Fe-AOM (Yu et al., 2022) or switch between methane generation and Fe reduction (Eliani-Russak et al., 2023; Sivan et al., 2016). In contrast, Fe reduction and potentially also $Fe^{2+}$ liberation can occur (largely) abiotically by reactions with inorganic compounds such as FeS or $FeS_2$ (Bottrell et al., 2010; Mortimer et al., 2011) and hydrogen sulfide (sulfide oxidation by reduction of Fe(III), e.g., Canfield, 1989; Holmkvist et al., 2011; Pyzik and Sommer, 1981; Riedinger et al., 2010; Thamdrup et al., 1993) as well as by reactions with organic molecules (e.g., oxalate), which themselves might be produced by microbial activity (e.g., Burdige, 1993; Ionescu et al., 2015, and references therein). Recently, Aromokeye et al. (2021) suggested that

Fe reduction in methanic sediments of the North Sea occurs concomitantly with the use of crystalline Fe-oxides as conduits for interspecies electron transfer between fermentative bacteria and methanogens (methanogenic benzoate fermentation). The mechanistic details of this process are still to be fully understood. Clearly, abiotic and biotic reactions of Fe in marine sediments are closely interrelated with each other. Moreover, all of them are directly or indirectly linked to the biogeochemical cycling of C and S. In order to fully assess these interlinks and in particular to determine their relevance for methane generation and/or consumption, we require a better understanding of deep Fe reduction pathways in natural settings and their dependence on environmental conditions. Differentiation between abiotic and biotic Fe reaction pathways – in particular in methanic environments – would furthermore be of interest for studying life in extreme environments such as the sedimentary deep biosphere, where the availability of degradable organic matter and electron acceptors yielding high standard free energies is often strongly limited (e.g., Heuer et al., 2017; D'Hondt et al., 2004). Here, we investigate whether combined pore-water and solid-phase stable Fe isotope signatures can be used to differentiate between biotic and abiotic Fe reduction pathways in methanic sediments. A similar approach for assessing the dominance of Fe–S reactions over MIR and vice versa using Fe isotopes was successfully applied in shallow sediments of the continental margin off California (Severmann et al., 2006). We also specifically investigate the Fe isotopic signals of crystalline Fe-oxides, including magnetite, because for the site investigated here, these minerals were found to stimulate deep Fe release based on their conductivity (Aromokeye et al., 2021).

To our knowledge, there have only been two studies so far focusing on Fe reduction in methanic sediments that also include pore-water Fe isotope data: Sivan et al. (2011) proposed the occurrence of Fe-AOM in sediments of Lake Kinneret (Israel) and showed a light isotopic composition of pore-water $Fe^{2+}$ ($\sim -2\,‰$) in the respective interval. A recent study on very old and compacted sediments of the Nankai Trough off Japan (IODP site C0023) showed that extremely negative pore-water $\delta^{56}Fe$ values of up to $-5.9\,‰$ are most likely derived from a combination of MIR and Rayleigh fractionation, where $^{56}Fe^{2+}$ is preferentially adsorbed onto mineral surfaces (Köster et al., 2023).

Despite the lack of data concerning Fe isotope fractionation during biotic reduction pathways other than DIR, we assume that microbially mediated Fe liberation in methanic sediments similarly results in a preferential release of $^{54}Fe^{2+}$ and, thus, shifts pore-water $\delta^{56}Fe$ towards negative values. Furthermore, we hypothesize that the microbial utilization of a specific Fe-(oxyhydr)oxide pool results in relative enrichment of $^{56}Fe$ in the remaining substrate, enhancement which is detectable by a combination of sequential solid-phase Fe extractions and $\delta^{56}Fe$ analyses after Staubwasser et al. (2006) and Henkel et al. (2016, 2018). We present a

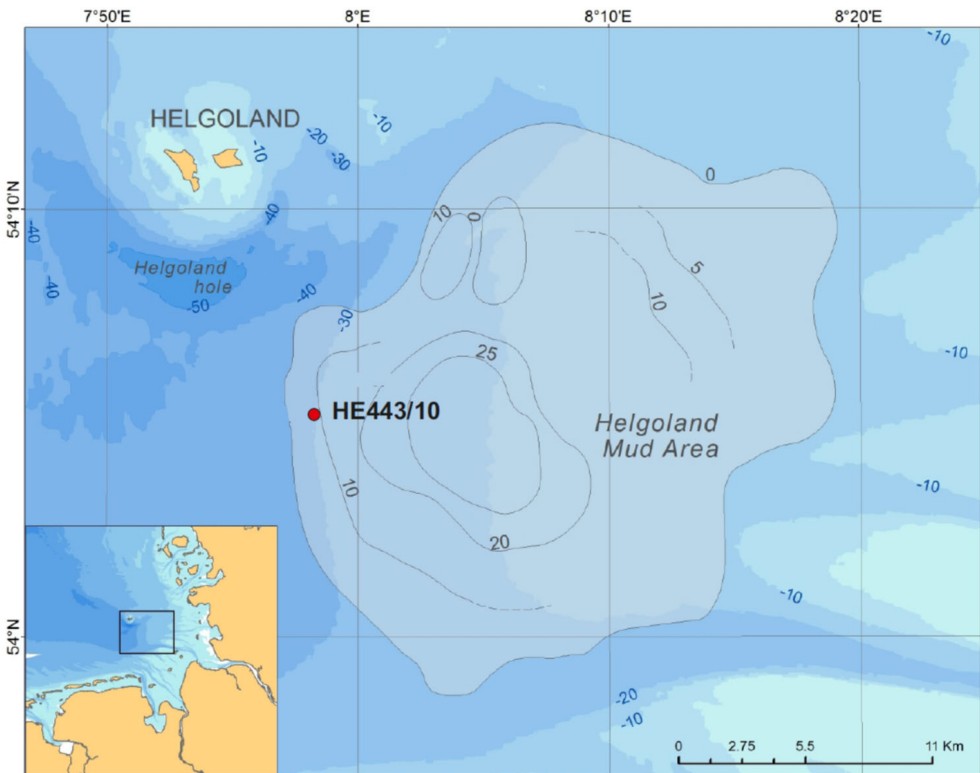

**Figure 1.** Position of cores collected during RV *Heincke* expedition HE443. Contour lines indicate the thickness of the mud in meters after Hebbeln et al. (2003).

compilation of inorganic geochemical data including pore-water and bulk solid-phase geochemistry, iron monosulfide and pyrite extractions, and $\delta^{56}$Fe in pore water and reactive Fe pools in order to assess sources and sinks of dissolved iron in the methanic sediments of the Helgoland mud area (HMA, German Bight, North Sea). We exemplarily tested if the $\delta^{56}$Fe data, in combination with geochemical transport-reaction modeling and the information from previous micro-biological studies, can be used to trace and explain the con-tributions of Fe reduction processes at different depths in the sediment column and to discriminate abiotic from biotic re-duction pathways.

## 2 Materials and methods

### 2.1 Study area

The Helgoland mud area (HMA, Fig. 1) is a halotectonic de-pression in the German Bight filled with Holocene mud that is mostly discharged by the rivers Elbe and Weser (e.g., Her-tweck, 1983; Irion et al., 1987). It is one of the few depocen-ters of fine-grained and organic-carbon-rich sediments in the German Bight, extends over $\sim 500\,\mathrm{km}^2$, and has an average water depth of 20 m (Hebbeln et al., 2003). Irion et al. (1987) explained that between 10 000 and 8000 BP the Elbe es-tuary was located at the present position of the HMA and that the old glacial relief formed a barrier towards the north that allowed the mud deposition in a comparatively protected bight. According to Irion et al. (1987), this protective barrier was destroyed at about 3000 to 2000 BP due to wave and tide activities. High sedimentation rates of more than $13\,\mathrm{mm\,yr^{-1}}$ characterized the HMA between $\sim 1250$–700 BP and were attributed to the disintegration of the island of Helgoland in the Middle Ages (Hebbeln et al., 2003). The reduction in sed-imentation rates to $< 3\,\mathrm{mm\,yr^{-1}}$ after $\sim 700$ BP was linked to a slowdown of the disintegration of the island and/or a change in the deposition location (Hebbeln et al., 2003). The high sedimentation rates that prevailed in the region of the HMA before 700 BP led to a fast burial and good preserva-tion of reactive Fe-(oxyhydr)oxides, which is a prerequisite of deep Fe reduction.

Previous investigations in the western have HMA shown a deep Fe release from sediments below the sulfidic zone. The studies by Oni et al. (2015a) and Aromok-eye et al. (2020, 2021) demonstrated that the Fe release is directly or indirectly related to microbial activity. Oni et al. (2015a) found a correlation of dissolved iron (Fe$_{\mathrm{diss}}$) concentrations with the abundance of Atribacterota (for-merly known as candidate phylum JS1), methanogens, and *Methanohalobium*/anaerobic methanotrophic archaea-3 (ANME-3)-related archaea. It was therefore suggested that

these microbes could be involved in deep Fe reduction, possibly via simultaneous occurrence of methanogenesis and Fe-mediated AOM. Aromokeye et al. (2020) performed an incubation experiment on sediment material from the same site as ours (parallel core, site HE443/10) and could detect Fe-AOM in slurries from the methanic zone, in particular in long-term incubations (250 d) amended with synthetic Fe-oxides. Dissolved Fe was, however, also released from sediments of the methanic zone when those were incubated with $N_2$ in the headspace (so unrelated to methane oxidation), irrespective of synthetic Fe-oxide addition. The authors therefore concluded that there are additional processes for deep $Fe_{diss}$ generation that are directly linked to organic matter (OM) degradation. One of these pathways was later suggested to be coupled to the fermentation of complex organic matter (Aromokeye et al., 2021).

## 2.2 Sediment and pore-water sampling

The data shown here are derived from a multiple-corer (MUC) deployment and a gravity core (GC) collected during RV *Heincke* cruise HE443 in April 2015 (MUC: 54°05.14′ N, 7°58.15′ E; GC: 54°05.19′ N, 7°58.21′ E; water depth 30 m, Fig. 1). The GC, site HE443/10-3, had a length of 488 cm. As surface sediment is usually lost during GC coring, the data of the MUC, site HE443/10-2, were used to fit GC pore-water data and calculate actual sediment depths for GC samples.

Directly after coring, the GC was cut into 1 m segments. Samples for methane ($CH_4$) analysis (3 mL of sediment) were taken immediately at the segment ends and stored in headspace vials that were pre-filled with 20 mL of a saturated NaCl solution containing $NaN_3$. The vials were sealed and stored at 4 °C until analysis. The GC segments were further sampled through small windows cut into the liner. First, additional $CH_4$ samples were collected. Then, pore water was extracted at 20 cm intervals using Rhizon samplers with an average pore size of 0.1 μm (Seeberg-Elverfeldt et al., 2005). The first $\sim$ 1.5 mL of collected pore water was discarded. Afterwards, 1.5 mL was collected and mixed with zinc acetate (ZnAc) for onshore sulfide measurements, 2 mL was stored in glass vials without headspace for dissolved inorganic carbon (DIC) analysis, 400 μL was diluted 1 : 10 with a NaCl solution for ammonium ($NH_4$) and phosphate ($PO_4^{2-}$) analyses, and $\sim$ 1 mL was stored for onshore sulfate ($SO_4^{2-}$) and chloride ($Cl^-$) measurements. After collection of pore water for the abovementioned parameters, new syringes pre-filled with 50 μL of concentrated double-distilled $HNO_3$ were attached to the Rhizon samplers and another 1–2 mL was collected for cation analysis. Pore-water aliquots were all stored at 4 °C. To maximize the sample volume, pore water for $\delta^{56}Fe$ analyses was collected in between the depths sampled for all other pore-water parameters, also at 20 cm intervals. However, we only processed every second to third sample and based the sample selection on the $Fe_{diss}$ profile shape with a higher resolution close to reaction fronts and

a lower resolution where $Fe_{diss}$ shows a rather linear gradient. Those samples were collected in pre-cleaned PP vials: for 1 d, 3 % ELMA 70 (an alkaline detergent); for 1 d, deionized water; for 7 d, 0.3 M HCl; and for 3 d, ultra-pure water. All $\delta^{56}Fe$ samples were acidified with double-distilled HCl. Solid-phase samples were collected using cutoff syringes, tightly sealed, and stored at $-20$ °C in Ar-filled CE1 gastight glass containers. The GC was cut open 6 months after coring, and X-radiographs were produced at the Alfred Wegener Institute, Helmholtz Centre for Polar and Marine Research (AWI) in Bremerhaven.

We collected three parallel MUCs during one deployment at the same location that was cored with the GC: one for pore-water $\delta^{56}Fe$ sampling, one for solid-phase analyses, and one for all other parameters mentioned above for the GC except for $CH_4$. Pore-water sampling was conducted as described above but at 1 cm intervals down to a depth of 10 cm and every 2 cm further below. Solid-phase samples were gained by slicing the MUC core at 1–2 cm intervals. These samples were all treated as previously described.

Both the MUC and the GC core, HE443/10-2 and HE443/10-3, consisted of dark to very dark grey mud. The GC that was cut open onshore showed bioturbation structures over the whole core length. Only at the following intervals (core depths) was sediment lamination still largely intact: 20 to 27 cm, 121 to 135 cm, 190 to 195 cm, 238 to 264 cm, 273 to 278 cm, 311 to 326 cm, 340 to 388 cm, and 441 to 447 cm. Layers and lenses of silty material are present at 60, 275, 350 to 360, and 370 to 380 cm. X-radiographs of the core are accessible in the PANGAEA database (Henkel et al., 2024a).

## 2.3 Pore-water analyses

The following pore-water analyses were conducted at the AWI: DIC, $NH_4$, and $PO_4$ were measured directly after the cruise using a QuAATro SEAL Analytical nutrient analyzer. The methods used were as described in the user handbook (Q-067-05, Q-080-06, Q-064-05) and are based on Stoll et al. (2001) (for DIC), Kerouel and Aminot (1997) (for $NH_4$), and the complex formation of $PO_4^{3-}$ with ammonium molybdate, respectively. The pH value was measured in sampled pore water using a pH electrode and a WTW pH meter. Cation concentrations (Fe, Mn, Ca, Mg) were determined by inductively coupled plasma–optical emission spectrometry (ICP-OES, IRIS Intrepid II). Dissolved sulfide ($\sum H_2S = H_2S + HS^- + S^{2-}$) was analyzed using the methylene blue method (Cline, 1969). Sulfate and chloride were measured in 1 : 50 dilution using a Metrohm 761 Compact IC ion chromatograph. Headspace gas for $CH_4$ analysis was injected into a Thermo Finnigan TRACE GC equipped with a packed column and an integrated flame ionization detector (FID). Methane concentrations were calculated under consideration of an average porosity of 0.7, determined based on the water

content of sediment samples and an estimated average grain density of $2.6\,\mathrm{g\,cm^{-3}}$.

Pore-water processing for $\delta^{56}$Fe analysis was conducted at the University of Cologne: Fe was first pre-concentrated and extracted from the salt matrix using the NTA Superflow resin as described by Henkel et al. (2016, 2018). Subsequently, samples were further purified by anion exchange chromatography using 150 µL Dowex 1X8 200-400 resin. All columns and vials used during the processing were pre-cleaned with ELMA 70 and diluted HCl as described above for the PP sampling vials. The purified Fe samples were matched to a concentration of 0.2 ppm and introduced into a Thermo Finnigan Neptune multicollector inductively coupled plasma–mass spectrometry (ICP-MS) instrument equipped with an Aridus desolvating nebulizer system at the Steinmann Institute in Bonn. We applied the standard-sample bracketing method with IRMM-014 (e.g., Schoenberg and von Blanckenburg, 2005). All $^{54}$Fe data were Cr-corrected based on measurements of $^{53}$Cr. In addition, all data were blank-corrected and samples were analyzed in random order. Data are reported as $\delta^{56}\mathrm{Fe}\,[\text{‰}] = (^{56}\mathrm{Fe}/^{54}\mathrm{Fe}_{\text{sample}})/(^{56}\mathrm{Fe}/^{54}\mathrm{Fe}_{\text{IRMM-014}}) - 1] \times 1000$. Details concerning the instrumental setup can be found in Henkel et al. (2016, 2018). We monitored the measurement trueness of the isotopic analyses by use of the reference material JM, a solution produced from an Fe wire supplied by Johnson Matthey. A JM was measured after each set of six samples. The measured value was $0.49 \pm 0.26\text{‰}$ ($n = 15$, 2 SD) and overlapped within uncertainty with previously published values ($0.42 \pm 0.05\text{‰}$, Schoenberg and von Blanckenburg, 2005; $0.46 \pm 0.20\text{‰}$, Walczyk and von Blanckenburg, 2005; $0.35 \pm 0.14\text{‰}$, Weyer and Schwieters, 2003). The uncertainty in a single measurement (2 SD of one block of 20 measurement cycles) was between 0.07 and 0.14 ‰. Processing and analysis of two duplicate samples resulted in $\delta^{56}$Fe values within the uncertainties (2 SD) of the respective single measurements. The trueness of the entire sample processing procedure was controlled by processing of blanks, the isotope standard IRMM-014, and Fe standards (Certipur®) in different concentrations. Blanks yielded Fe concentrations of 3 ppb, a level which was at most 4 % of the Fe concentration in the pore-water samples. The processing of Certipur® Fe standards showed a recovery of > 85 % of Fe. Pore-water $\delta^{56}$Fe values were analyzed using a Keeling plot, which is traditionally used for carbon isotope mixing (Keeling, 1958; Pataki et al., 2003). Details are given in Sect. 4.2.

## 2.4 Solid-phase analyses

The bulk elemental composition of the sediment was determined by total digestion of about 50 mg of freeze-dried and ground sediment in a mixture of concentrated acids (3 mL HCl, 2 mL HNO$_3$, and 0.5 mL HF). The digestion was carried out in a CEM MARSXpress microwave system at AWI.

After evaporation of the acids, the residue was dissolved in 1 M HNO$_3$ and measured by ICP-OES. Recoveries of processed NIST SRM 2702 reference material ($n = 5$, uncertainty given as 2 SD) were $100.2 \pm 0.8$ % for Fe, $98.1 \pm 2.4$ % for Mn, $101.5 \pm 2$ % for Ca, and $93.8 \pm 2.8$ % for Al. Total Fe contents were published by Aromokeye et al. (2020) and are available in PANGAEA (Aromokeye et al., 2018a).

Sequential Fe extractions were performed after Poulton and Canfield (2005): $\sim 50$ mg of dry sediment was suspended in 5 mL of (a) MgCl$_2$ for adsorbed Fe, (b) Na-acetate for Fe-carbonates and surface-reduced Fe(II), (c) hydroxylamine–HCl for easily reducible Fe-oxides (ferrihydrite, lepidocrocite), (d) Na-dithionite/Na-citrate for reducible Fe-oxides (goethite and hematite), and (e) ammonium oxalate/oxalic acid for magnetite. In comparison to the method by Poulton and Canfield (2005), we used a lower concentration of citrate for the dithionite extraction since citrate hinders Fe precipitation during subsequent sample purification for $\delta^{56}$Fe analysis (Henkel et al., 2016). Instead, we performed the extractions under anoxic conditions. Aliquots of all extracts were analyzed by ICP-OES. A separate aliquot of 2 mL of each extract was processed for $\delta^{56}$Fe analysis following the protocol by Henkel et al. (2016). Henkel et al. (2016) demonstrated that the processing of samples does not lead to iron fractionation. The purified Fe samples were matched to a concentration of 0.5 ppm and were analyzed using the Thermo Fisher Scientific Neptune Plus MC ICP-MS instrument of the Isotope Geochemistry Group at MARUM – Center for Marine Environmental Sciences, University of Bremen. The MC ICP-MS instrument was equipped with an SSI dual cyclonic spray chamber, a low-flow 50 µL PFA nebulizer, and a Ni skimmer cone (X-type). Samples were measured using the standard-sample bracketing with certified reference material IRMM-014. All $^{54}$Fe data were Cr-corrected based on measurements of $^{52}$Cr. In addition, all data were blank-corrected and samples were analyzed in random order. The standard JM (see above) was analyzed after each block of three samples. Samples bracketed by JMs that did not fall into the target range of $0.42 \pm 0.05$ ‰ were repeatedly measured. The repeatability precision resulting from up to six replicate sample measurements (not including replicate processing) was in the worst case 0.34 ‰ (2 SD) and in the best case 0.03 ‰ (average 0.11 ‰; see Fig. 5). The intermediate precision of JMs was $0.44 \pm 0.15$ ‰ ($n = 151$, 2 SD).

Acid volatile sulfide (AVS; mostly iron monosulfides) and chromium reducible sulfide (CRS; mostly pyrite but potentially also elemental sulfur) were extracted at AWI by hot digestion using 6 M HCl and a chromous chloride solution, respectively (Canfield et al., 1986; Praharaj and Fortin, 2003; Wieder et al., 1985). Extracted sulfur was trapped in a silver nitrate solution as Ag$_2$S. After filtration, the dry masses of the precipitates were converted into FeS and FeS$_2$ contents based on stoichiometry. Replicate analysis of an in-house standard (core catcher sediment of GC HE443/077) revealed good reproducibility of the extractions with AVS contents

of $0.11 \pm 0.01$ wt % and pyrite contents of $1.03 \pm 0.05$ wt % ($n = 7$).

On freeze-dried, powdered, and homogenized sediment samples, the total carbon (TC) contents were determined using a CNS (Elementar vario EL III) analyzer. Total organic carbon (TOC) contents were measured with a carbon–sulfur analyzer (CS-2000, ELTRA) after removal of inorganic carbon with HCl.

## 2.5   Model setup and parameterization

A reactive transport model was used in order to (1) exemplarily assess if the measured pore-water Fe and $\delta^{56}$Fe profiles at site HE443/10 can be reproduced based on Fe reactions that are known to occur (MIR and Fe sulfide formation) and (2) delineate how sensitive the pore-water Fe and $\delta^{56}$Fe profiles are with respect to different reaction rate constants $k$ and related fractionation factors. To keep this approach as simple and straightforward as possible, we only included the most basic and presumably dominant reactions that are known to affect the dissolved Fe pool and its isotopic composition: organoclastic DIR, reaction of hydrogen sulfide with Fe(III) to Fe sulfide, and the precipitation of Fe sulfides by counter-diffusion of pore water $Fe^{2+}$ and $HS^-$ (Reactions R1–R3, Table S1 in the Supplement).

$$CH_2O + 4Fe(OH)_3 \rightarrow HCO_3^- + 4Fe^{2+} + 3H_2O + 7OH^- \quad (R1)$$

$$Fe(OH)_3 + 2HS^- \rightarrow FeS_2 + 2OH^- + H_2O + 0.5H_2 \quad (R2)$$

$$Fe^{2+} + 2HS^- \rightarrow FeS_2 + H_2 \quad (R3)$$

We are aware that we miss some reactions in the model that might play a role as well, e.g., siderite or vivianite precipitation, Fe-AOM, and reoxidation of sulfide by Fe(III). Furthermore, Reaction (R3) is actually not a single reaction but includes the formation of monosulfide (FeS) and (in a second step) the transformation of FeS into $FeS_2$, where the latter reaction can happen abiotically but can also be driven by microbes (Thiel et al., 2019). Our approach is basically backwards as we check whether we can reproduce the profile shapes of dissolved Fe and the respective $\delta^{56}$Fe values from the deep Fe source to the sink at the sulfidization front sufficiently well by just including these basic reactions or whether we miss a reaction that would be needed to explain the measurements. With regard to siderite and vivianite formation, a calculation with PHREEQC (see Sect. 2.6) in fact indicates oversaturation below the SMT at site HE443/10-3. Nevertheless, we chose to neglect these reactions in the model as the specific contributions are unclear (siderite and vivianite) or respective Fe fractionation factors are unknown (vivianite). We discuss, however, how particularly siderite precipitation could affect our results, e.g., Fe extraction data and the dissolved Fe isotope profile in Sect. 4.1 and 4.2. Since we were primarily interested in the deep Fe reduction, modeling was confined to the sediment interval between 70 cm (sulfide peak) and 450 cm (end of core). We disregarded all the reduction and oxidation processes above the sulfide peak as they are irrelevant for the expression of aqueous $\delta^{56}$Fe below the sulfidic zone. This is because $Fe^{2+}$ is completely removed within the sulfidic zone due to the reaction with $HS^-$.

The reaction rates were obtained according to the concentrations of the reacting species. For example, Reactions (R2) and (R3) approach zero when $HS^-$ is depleted. The following transport-reaction equations for $Fe^{2+}$ and $HS^-$ were used:

$$\frac{\partial[Fe^{2+}]}{dt} = -\omega \frac{\partial[Fe^{2+}]}{\partial z} + \frac{D_{Fe^{2+}}}{\tau^2} \cdot \frac{\partial^2[Fe^{2+}]}{\partial z^2} + 4R_1 - R_3, \quad (1)$$

$$\frac{\partial[HS^-]}{dt} = -\omega \frac{\partial[HS^-]}{\partial z} + \frac{D_{HS^-}}{\tau^2} \cdot \frac{\partial^2[HS^-]}{\partial z^2} + 2R_2 - 2R_3, \quad (2)$$

where $[Fe^{2+}]$ and $[HS^-]$ are the concentrations of dissolved iron and sulfide, $t$ is time, $\omega$ is the sedimentation rate, $z$ is the depth below seafloor, and $D_{Fe^{2+}}$ and $D_{HS^-}$ are the diffusion constants for dissolved iron and sulfide. The applied sedimentation rate ($0.16$ cm yr$^{-1}$) is derived from Hebbeln et al. (2003). Diffusion constants for seawater at pore-water temperature ($4\,^\circ$C) are from Boudreau (1996b) ($D_{Fe^{2+}} = 0.0116$ m$^2$ yr$^{-1}$ and $D_{HS^-} = 0.0306$ m$^2$ yr$^{-1}$). A constant porosity ($\varphi$) of 0.7 was assumed, and the tortuosity ($\tau$) in Eqs. (1) and (2) was calculated according to Boudreau (1996a) as $\tau^2 = 1 - \ln(\varphi^2)$. Although the OM degradation rate constant $k_1$ decreases with burial depth (Arndt et al., 2013), its variation is low below the SMT under high-burial-rate conditions. For simplicity, the Fe reduction rate coupled with OM degradation is assumed to follow the first-order decay model (E1; see Table S1). The pool of reducible Fe-oxides is set to not be limiting and is based on the data gained from sequentially extracted Fe, and its $\delta^{56}$Fe composition was kept at a constant value of $0.0\permil$ (see Results). Within the sulfidic zone there is no free $Fe^{2+}$ (assumption: $0.01\,\mu$M at the upper boundary), and all $Fe^{2+}$ released from the abiotic reaction with $HS^-$ (sulfide oxidation by reduction of iron (oxyhydr)oxides) is assumed to be immediately converted into pyrite (Reaction R2). Due to the complete turnover of released $Fe^{2+}$, it is reasonable to assume that there is no related isotopic fractionation ($\alpha_2 = 1.000$). During Fe sulfide formation where $Fe^{2+}$ and $HS^-$ counter-diffuse, we applied the kinetic fractionation factor $\alpha_3$ ($\alpha_{Fe_{py}-Fe(II)_{diss}}$), which was set to 0.999, 0.998, and 0.997 to fit the measured data and resulted in dissolved Fe with an isotopic composition $\delta^{56}Fe_{diss} \sim 0\permil$ compared to more negative values presented below (see Results and Discussion), where Fe concentrations are higher (upward Fe diffusion). In other words, Fe sulfide formation via the reaction of $Fe^{2+}$ with $HS^-$ preferentially incorporates $^{54}$Fe (e.g., Butler et al., 2005; Scholz et al., 2014; Severmann et al., 2006). In addition, in different model runs we applied a fractionation factor $\alpha_1$ ($\alpha_{Fe(II)_{diss}-Fe(III)}$) = 0.998, 0.997, and 0.996, for DIR below 70 cm depth in order to reproduce the measured pore-water $\delta^{56}$Fe profile. These values are in the range of DIR fractionation factors published by Beard et al. (1999, 2003b), Crosby et al. (2007), Johnson et al.

(2005), and Severmann et al. (2006). It is important to note that the factors we apply do not resolve all the partial fractionation processes involved but only the fractionation related to the sum of Reactions (R1) and (R3). Our $\alpha_1$ for example reflects the isotopic difference between solid-phase Fe(III) and dissolved Fe(II), but in reality isotopic fractionation happens not only during the microbial Fe reduction and $Fe^{2+}$ release, but also between adsorbed Fe(II) and a reactive solid Fe(III) and adsorbed Fe(II) and the dissolved Fe(II), respectively (Crosby et al., 2005, 2007). Furthermore, we note that it is a valid approach to use $k$ as a fitting parameter because for biotic reactions, the rate constant depends not only on temperature, but also on the abundance and activity of microbes. Typically, this leads to a very large range of constant values. For example, the rate constant was given as 100 and $14\,800\,\mathrm{mM^{-1}\,yr^{-1}}$ for the same reaction, $Fe^{2+} + H_2S \rightarrow FeS$, in Reed et al. (2011a) and Reed et al. (2011b), respectively. In our study, $k_2$ also depends on the contents of reactive $Fe(OH)_3$ because Reaction (R2) is generally expressed as $k[Fe(OH)_3][H_2S]$. $k_3$ combines the fast FeS formation rate constant $k[Fe^{2+}][H_2S]$ and slow $FeS_2$ rate constant $k[FeS][H_2S]$. All model parameters and boundary conditions are given in Table S2 in the Supplement.

## 2.6 Calculation of saturation indices (SIs)

The saturation indices of selected secondary Fe minerals, namely vivianite and siderite, were calculated using the computer program PHREEQC (Parkhurst and Appelo, 2013). The thermodynamic database "phreeqc.dat" was used because it has a relatively wide range of aqueous complexation reactions for 25 chemical elements, including P and Fe. The input files defined for the geochemical calculations in PHREEQC are based on measured DIC; pH; and the aqueous concentrations of $Mg^{2+}$, $PO_4^{3-}$, $NH_4^+$, $SO_4^{2-}$, $HS^-$, $Cl^-$, $Mn^{2+}$, and $Fe^{2+}$. $NO_3^-$ was set to zero as it is already depleted close to the sediment surface. Since the redox potential ($E_h$) is a mandatory input variable for these types of geochemical calculations but was unavailable, the default values in PHREEQC were used. The effect of $E_h$ on the saturation of vivianite and siderite was insignificant as determined by a sensitivity test. The in situ temperature of the pore water was set to 4 °C. The concentration of $Fe^{2+}$ within the sulfidic zone was set to 1 µM as the detection limit.

## 3 Results

Based on the DIC and $SO_4^{2-}$ profiles of the MUC and GC cores, the core loss during gravity coring was determined to be 16 cm. Core depths of the GC were corrected to sediment depth accordingly.

## 3.1 Pore-water geochemistry

The pore-water profiles of $SO_4^{2-}$, $HS^-$, $CH_4$ (only end of core segments), DIC, and dissolved Fe and Mn of GC HE443/10-3 were shown earlier in Aromokeye et al. (2020) and are available in the PANGAEA database (Aromokeye et al., 2018b; core depths instead of sediment depths).

The pore-water profiles at site HE443/10 indicate ferruginous conditions at 1–2 cm depth. Dissolved Fe concentrations peak at 5 cm with 180 µM and then decrease towards 10 cm depth, where Fe and dissolved sulfide counter-diffuse (Fig. 2). Based on concurrent analysis of $Fe^{2+}$ (using the ferrozine method after Stookey, 1970) and dissolved Fe with ICP-OES on several other sites of the same expedition, we are confident that all dissolved Fe at station HE443/10 is in the form of $Fe^{2+}$. Sulfate shows a kink-shaped profile with a minor decrease from the sediment–water interface to 15 cm depth (25.3 to 24.3 mM) and a steeper gradient further downcore to $\sim 0$ mM at 86 cm. Sulfidic conditions prevail between 10 and 100 cm depth. Sulfide concentrations in the pore water peak at $\sim 70$ cm depth (0.5 mM), where $SO_4^{2-}$ and $CH_4$ counter-diffuse. Right below the SMT, $CH_4$ concentrations increase to $\sim 3$ mM and more. Higher values, $\geq 4$ mM, were measured in samples directly taken from ends of core segments during the cutting of the core. At depth, $CH_4$ concentrations do not significantly increase. Below the sulfidic zone, dissolved Fe ($Fe_{diss}$) concentrations gradually increase downcore to 400 µM at 350 cm. The concentrations remain at this level further below. The $\delta^{56}Fe_{diss}$ profile shows the most negative values at the sediment–water interface ($-1.8\,‰$) (Fig. 2). The values increase to about $-1\,‰$ at 8 cm depth. There are no $\delta^{56}Fe_{diss}$ values for the sulfidic zone due to absence of $Fe_{diss}$. Right below the sulfidic zone, where $Fe_{diss}$ concentrations of $\sim 100$ µM were detected, $\delta^{56}Fe_{diss}$ is $0\,‰$. As $Fe_{diss}$ concentrations increase further downcore, there is a shift to $-1.3\,‰$ at 186 cm followed by a gradual increase in values towards $0\,‰$ at 450 cm, where $Fe_{diss}$ peaks.

Phosphate concentrations show an increase from 9 µM at 1 cm depth to $\sim 530$ µM at 90 cm. Concentrations then decrease gradually to $\sim 250$ µM at depth (Fig. 3). Phosphorus concentrations measured by ICP-OES of acidified pore-water aliquots (not shown but available under Henkel et al., 2024a) mirror the overall $PO_4$ profile so that we can exclude a drawdown of $PO_4$ at depth as a sampling artifact in Fe-rich pore water from below the SMT. Oxidation of samples easily leads to Fe precipitation and $PO_4$ drawdown due to adsorption. The Mn pore-water profile shows concentrations of $\sim 50$ µM at 3 cm sediment depth. Towards the sulfidic zone, concentrations decrease to zero. A second maximum of Mn concentrations ($\sim 200$ µM) is located at 200 cm. Down to this depth, the Mn profile shape mirrors the $Fe_{diss}$, although concentrations are considerably lower. Unlike $Fe_{diss}$, Mn concentrations then decrease to 15 µM at 415 cm. Towards the end of the core, Mn increases again to 40 µM. Dissolved Ca concentrations show an overall downcore decrease from 9 to 5 mM

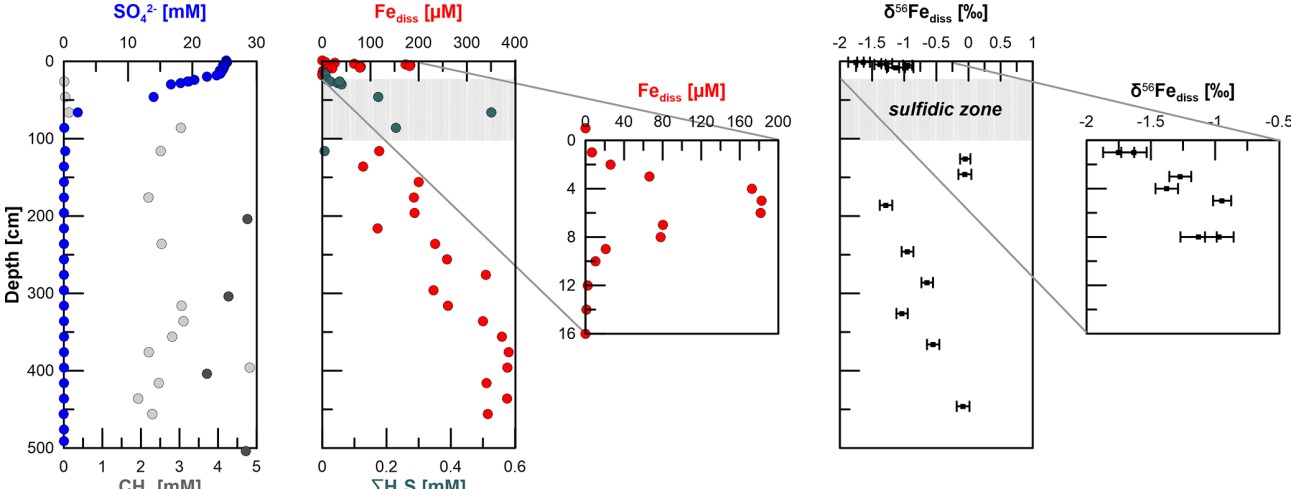

**Figure 2.** Pore-water profiles of $SO_4^{2-}$ and $CH_4$ at station HE443/10. Dark grey dots of the $CH_4$ graph indicate samples from ends of core segments. Those concentrations are more reliable compared to the others, as samples were directly taken during cutting of the core segments. The second and third panels show dissolved Fe and $H_2S$ concentrations as well as the stable Fe isotope values of dissolved Fe in the non-sulfidic sediments (uncertainty bars are 2 SD determined for the 20 measurement cycles in one "block" of analysis). Sulfate, methane (only end of core segments), sulfide, and dissolved-iron data of the gravity core were published earlier by Aromokeye et al. (2020). In all plots, the grey-shaded area indicates the sulfidic interval.

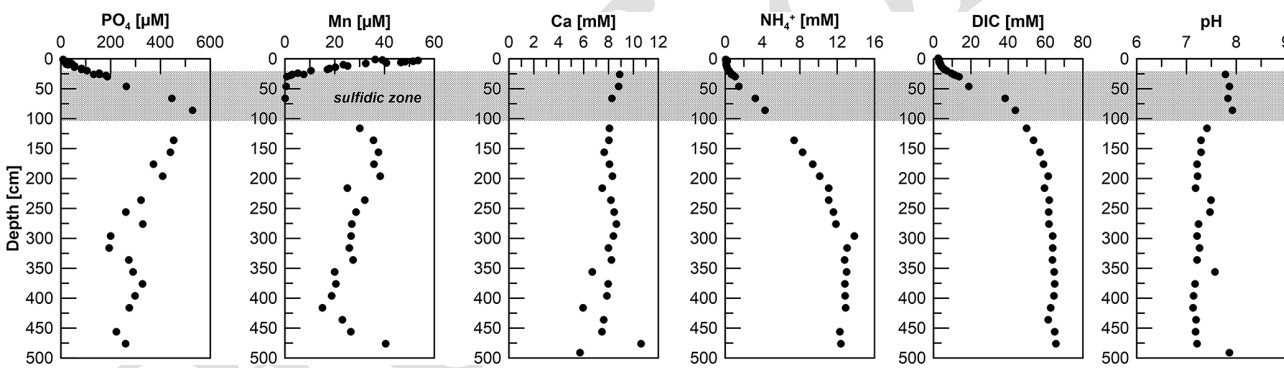

**Figure 3.** Pore-water $PO_4$, Mn, Ca, $NH_4^+$, DIC, and pH data for site HE443/10. The Mn pore water profile of the gravity core was published earlier by Aromokeye et al. (2020).

(Fig. 3). Ammonium concentrations show a steady increase from 45 µM at 1 cm depth to 13.8 mM at 300 cm. The concentrations remain at this level for the rest of the core (Fig. 3). The pH averages are at 7.85 above and at 7.29 below the SMT, and DIC increases from 2.5 mM in the bottom water to 66 mM at 476 cm sediment depth (Fig. 3).

## 3.2 Solid-phase composition

Total Fe contents ($Fe_{total}$) of GC HE443/10-3 were shown earlier by Aromokeye et al. (2020) and are available in the PANGAEA database (Aromokeye et al., 2018a; with core depths instead of sediment depth). Sequentially extracted Fe data as well as Mn and Al contents are available in PAN-GAEA (Henkel et al., 2024b).

$Fe_{total}$ varies between 17 and $42\,\mathrm{mg\,g^{-1}}$ (Fig. 4). The Fe/Al ratio ($\mathrm{g\,g^{-1}}$) at site HE443/10 is between 0.49 and 0.69 (average 0.59), with higher Fe/Al values corresponding to high $Fe_{total}$ contents. According to the sequential extraction data, 16 %–30 % of $Fe_{total}$ is associated with Fe-carbonates, FeS (which is not targeted here but dissolves in 1 M Na-acetate and was targeted in a separate extraction of Fe sulfides; see below), and Fe-oxides. There is no clear downcore decrease in $Fe_{total}$ or the sequentially extracted Fe pools (Fig. 4). On the contrary, there are intervals of elevated Fe contents at 230–300 cm and below 400 cm, which are reflected by all extracted Fe phases. Only when plotted relative to reactive Fe (sum of Fe extracted by $MgCl_2$, Na-acetate, hydroxylamine–HCl, Na-dithionite/Na-citrate, and ammonium oxalate/oxalic acid), the acetate-leached Fe pool (Fe-carbonates and surface-reduced Fe(II)) shows an overall

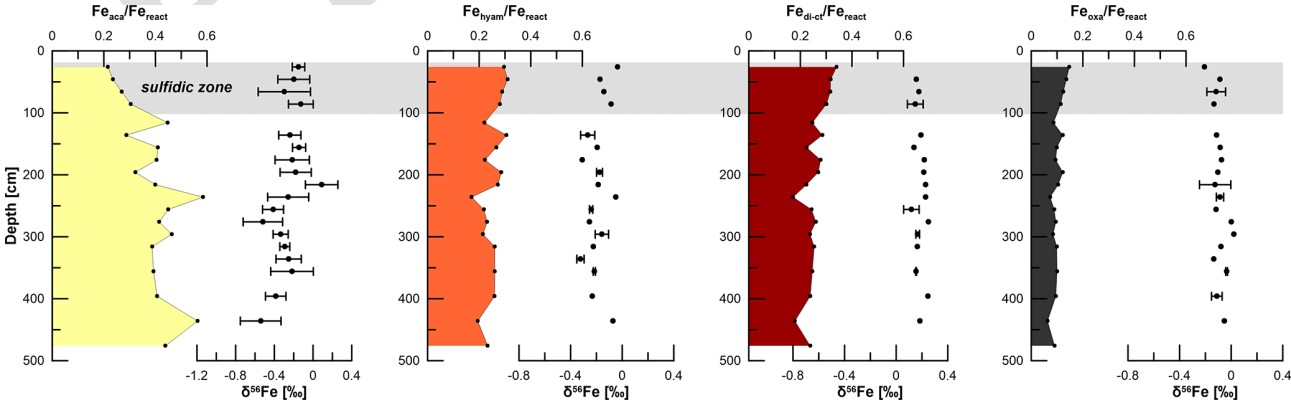

**Figure 4.** Total and reactive Fe contents, Fe/Al, Mn, AVS–Fe, CRS–Fe, and sequentially extracted Fe pools after Poulton and Canfield (2005). Note that reactive Fe is the sum of $Fe_{aca}$ (carbonates, surface-reduced Fe), $Fe_{hyam}$ (amorphous easily reducible oxides), $Fe_{dith}$ (goethite, hematite), and $Fe_{oxa}$ (magnetite). Note that the sequential extraction is not mineral-specific but operationally defined.

**Figure 5.** Sequentially extracted Fe pools normalized to the sum of reactive Fe and the respective $\delta^{56}Fe$ values. Uncertainty bars are 2 SD and given only for samples that were repeatedly analyzed. The grey bar indicates the sulfidic interval.

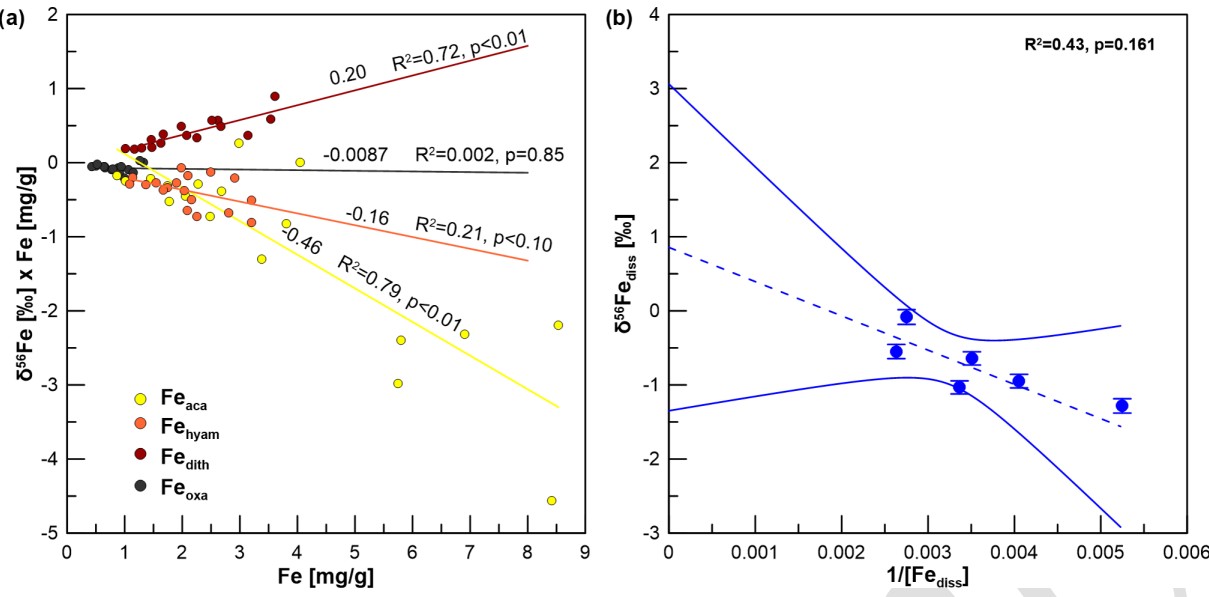

**Figure 6. (a)** Miller–Tans plot for $\delta^{56}$Fe values of sequentially extracted reactive Fe pools. **(b)** Keeling plot for $\delta^{56}$Fe values of pore water with the 95 % confidence interval. We only used data from between 450 and 150 cm, where there is a rather linear $\delta^{56}$Fe$_{diss}$ trend.

increase with sediment depth, whereas the other extracted Fe fractions rather show a decrease (Fig. 5). Close to the sediment surface, the composition of reactive Fe is as follows: 20 % acetate-leached Fe (Fe$_{aca}$), 30 % hydroxylamine–HCl-leached Fe (Fe$_{hyam}$), 35 % dithionite-leached Fe (Fe$_{di-ct}$), and 15 % oxalate-leached Fe (Fe$_{oxa}$). At 450 cm, the respective values are 50 % Fe$_{aca}$, 20 % Fe$_{hyam}$, 20 % Fe$_{di-ct}$, and 10 % Fe$_{oxa}$. Only the Fe$_{aca}$ pool shows a clear variation in the $\delta^{56}$Fe composition, where high Fe$_{aca}$ contents correspond to low $\delta^{56}$Fe$_{aca}$ signals down to $-0.54‰$ (Fig. 5). Data representation in a Miller–Tans plot (Miller and Tans, 2003; Fig. 6a), which allows assessing even small isotopic variations that depend on the size of the respective pool, underlines this relationship with $R^2 = 0.79$ ($p < 0.01$). The hydroxylamine–HCl-leached pool shows overall negative $\delta^{56}$Fe$_{hyam}$ values ($-0.19 \pm 0.17‰$, 2 SD). As with the Fe$_{aca}$ pool, the Miller–Tans analysis indicates a correlation between Fe$_{hyam}$ content and the isotopic composition (Fig. 6a), with higher contents being related to more negative $\delta^{56}$Fe$_{hyam}$ values. The relationship is, however, less clear ($R^2 = 0.21$, $p < 0.1$), and the overall range of $\delta^{56}$Fe$_{hyam}$ ($-0.32‰$ to $-0.03‰$) is considerably smaller compared to Fe$_{aca}$. $\delta^{56}$Fe$_{di-ct}$ values (Fig. 5) range between 0.12‰ and 0.25‰ ($0.19 \pm 0.08‰$, 2 SD). The Miller–Tans plot indicates that there is a slight enrichment of $^{56}$Fe in samples that show high Fe$_{di-ct}$ contents ($R^2 = 0.72$, $p < 0.01$) (Fig. 6a). Oxalate-leached Fe shows $\delta^{56}$Fe$_{oxa}$ values of $-0.09 \pm 0.10‰$ (2 SD) and neither a downcore trend (Fig. 5) nor a dependency of $\delta^{56}$Fe$_{oxa}$ on Fe$_{oxa}$ contents (Fig. 6a).

Total Mn contents in the solid phase range from 0.3 to 1 mg g$^{-1}$, where the overall profile shape is very similar to the distribution of Fe$_{total}$: maximum values occur between

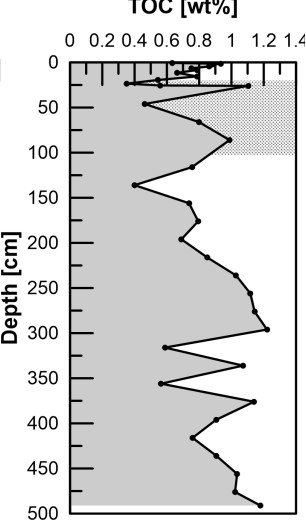

**Figure 7.** TOC in sediments from site HE443/10. The grey bar indicates the sulfidic interval.

200 and 300 cm. Local maxima further downcore coincide with maxima in Fe at 335, 375, 435, and 490 cm. TOC values range between 0.4 wt % and 1.2 wt % (Fig. 7) and show a strong positive correlation with Fe$_{total}$ ($R^2 = 0.76$, $p < 0.01$). As for Fe$_{total}$, there is no decrease in TOC with depth but a zone of elevated values between 230 and 300 cm and a trend towards higher values (up to 1.2 wt %) towards the end of the core. Total inorganic carbon (TIC; data available under Henkel et al., 2024b) shows similar trends, but values are higher (1.3 wt % to 2.3 wt %) compared to TOC.

Iron sulfides are present as AVS and CRS. Both sulfide pools are not limited to the sulfidic zone (Fig. 4) but occur over the whole gravity core length. AVS peaks at the depth of the current sulfidization front at $\sim 100\,\mathrm{cm}$ depth ($1\,\mathrm{mg\,g^{-1}}$ AVS–Fe). A second maximum of $2\,\mathrm{mg\,g^{-1}}$ AVS–Fe appears at $250\,\mathrm{cm}$. CRS is highest at $26\,\mathrm{cm}$ depth ($10\,\mathrm{mg\,g^{-1}}$ CRS-bound Fe), the uppermost sample that was analyzed. The contents decrease towards $\sim 120\,\mathrm{cm}$ ($3\,\mathrm{mg\,g^{-1}}$ CRS-bound Fe) and remain at a level of $\sim 4\,\mathrm{mg\,g^{-1}}$ further downcore. AVS is affected by the extraction of Fe-carbonates with Na-acetate (Cornwell and Morse, 1987; Poulton and Canfield, 2005). Based on the separate extraction, the contribution of AVS to the $\mathrm{Fe_{aca}}$ pool was calculated to be between 6 % and 26 % (average 13 %). This is particularly important with respect to $\delta^{56}\mathrm{Fe_{aca}}$ data. Even though there is no correlation between the AVS contribution to $\mathrm{Fe_{aca}}$ (in %) and the respective $\delta^{56}\mathrm{Fe_{aca}}$ value ($R^2 = 0.03$), $\delta^{56}\mathrm{Fe_{aca}}$ should be interpreted with caution as it represents a mixed signal of several (secondary) Fe pools that likely have different isotopic compositions: siderite, Fe monosulfides, and surface-reduced Fe(II) (Crosby et al., 2005, 2007; Henkel et al., 2016).

### 3.3 Model results (transport-reaction simulation)

We performed sensitivity tests by applying different fractionation factors $\alpha_1$ for DIR (Reaction R1, $k_1$ as a function of OM degradation), different reaction rate constants $k_2$ and a fractionation factor $\alpha_2 = 1$ for the sulfidization via the reaction of hydrogen sulfide with Fe-oxides (Reaction R2), and different reaction rate constants $k_3$ and fractionation factors $\alpha_3$ for sulfide precipitation via reaction of $\mathrm{Fe_{diss}}$ with $\mathrm{HS^-}$ (Reaction R3). The differences between the measured and calculated concentrations/values $C^i_{\mathrm{mea}}$ and $C^i_{\mathrm{calc}}$ at each depth $i$ were calculated using the mean square error (MSE) as

$$\mathrm{MSE} = \frac{1}{N} \sum_{i=1}^{N} \left( \frac{C^i_{\mathrm{mea}} - C^i_{\mathrm{calc}}}{C^i_{\mathrm{mea}}} \right)^2. \tag{3}$$

The minimized sum of the MSE for $\mathrm{Fe^{2+}}$, $\mathrm{HS^-}$, and $\delta^{56}\mathrm{Fe_{diss}}$ was used to find the best-fitting parameters. The respective constants $k_2$ applied for Reaction (R2) were 0.2, 0.4, and 0.8. The best fit based on the minimum MSE was achieved with $k_2 = 0.4$ (Fig. 8). The reaction front of $\mathrm{Fe_{diss}}$ and $\mathrm{HS^-}$ varies with the value $k_2$. Although there is no isotopic Fe fractionation considered for Reaction (R2), the changed $\mathrm{H_2S}$ profile leads to a different depth for Reaction (R3) and, thus, a different $\delta^{56}\mathrm{Fe_{diss}}$ profile. The constants $k_3$ tested for Reaction (R3) (in combination with $k_2 = 0.4$) were 2, 4, and 8. The $\mathrm{H_2S}$ concentration profile shows a higher dependency on Reaction (R2) (or $k_2$) compared to Reaction (R3) (or $k_3$). The best data fit resulted from applying $k_3 = 4$.

Using $k_2 = 0.4$ and $k_3 = 4$ for Reactions (R2) and (R3), the best fit of the modeled $\delta^{56}\mathrm{Fe_{diss}}$ profiles with the respective measured data was achieved when setting the kinetic fractionation factor $\alpha_1$ for MIR (DIR) (Reaction R1) to 0.997 and $\alpha_3$ for sulfide precipitation via reaction of $\mathrm{Fe^{2+}}$ with $\mathrm{HS^-}$ (Reaction R3) to 0.998 (Fig. 9). As Reaction (R1) is not confined to a narrow interval as is the case for Reaction (R3), the effect of the choice of $\alpha_1$ on the overall $\delta^{56}\mathrm{Fe_{diss}}$ profile is much stronger than the choice of $\alpha_3$. The value chosen for $\alpha_3$ is, of course, most relevant to the depth of $\sim 100\,\mathrm{cm}$, where $\mathrm{HS^-}$ formation occurs. Here, the choice of the fractionation factor results in differences in $\delta^{56}\mathrm{Fe_{diss}}$ of more than 2 ‰ (Fig. 9; modeled profile with $\alpha_3 = 0.997$ not shown completely due to the limitation of the $x$ axis to 1 ‰).

## 4 Discussion

### 4.1 Redox zones, reaction fronts, and sediment composition

The kink shape of the sulfate profile indicates some bioturbation and/or bioirrigation in the top $15\,\mathrm{cm}$ of the sediment (e.g., Henkel et al., 2011; Fischer et al., 2012), which is consistent with the general intense bioturbation at site HE443/10 evidenced by the radiographs (Henkel et al., 2024a). However, this does not result in a high $\mathrm{O_2}$ penetration depth as demonstrated by the presence of dissolved Mn ($\sim 35\,\mu\mathrm{M}$) at $1\,\mathrm{cm}$ and the maximum concentration at $3\,\mathrm{cm}$ (Fig. 3). Manganese oxides are considered to become microbially reduced as soon as the pore water is depleted of more favorable electron acceptors such as $\mathrm{O_2}$ and nitrate (e.g., Burdige, 1993). The $\mathrm{Fe_{diss}}$ profile (Fig. 2) is smooth and shows comparatively high concentrations of up to $180\,\mu\mathrm{M}$ at $5\,\mathrm{cm}$ depth. Therefore, we consider the direct effect of bioturbation and bioirrigation on pore-water geochemistry to be minor. The Fe liberation in this interval is mainly due to MIR, as has been described in detail in Henkel et al. (2016) for a site located $\sim 100\,\mathrm{m}$ away from site HE443/10.

The investigated sediments are rich in total and reactive Fe. The averages are 28 and $9\,\mathrm{mg\,g^{-1}}$ of sediment, respectively (Fig. 4). The Fe/Al ratio is $\sim 0.59$, which is close to the average shale composition of 0.55 (Wedepohl, 1991). There is no downcore decrease in $\mathrm{Fe_{total}}$, Fe/Al, or the reactive/extractable Fe pool as seen for example in Thamdrup et al. (1994) in sediments of the Bay of Aarhus (Denmark) and Severmann et al. (2006) in deposits of the Santa Barbara Basin. In the case of constant accumulation rates and a consistent composition of the accumulated material, a decrease in Fe or reactive Fe phases with depth is indicative of microbial Fe reduction, upward $\mathrm{Fe^{2+}}$ diffusion, and Fe-oxide precipitation at the redox boundary. The absence of such a decrease at site HE433/10 is likely due to intense reworking of the sediment. Bioturbation results in very effective mixing of solid phases in the top few centimeters.

Under the assumption that the Fe/Al ratio of the detrital material transported to the HMA area has been constant

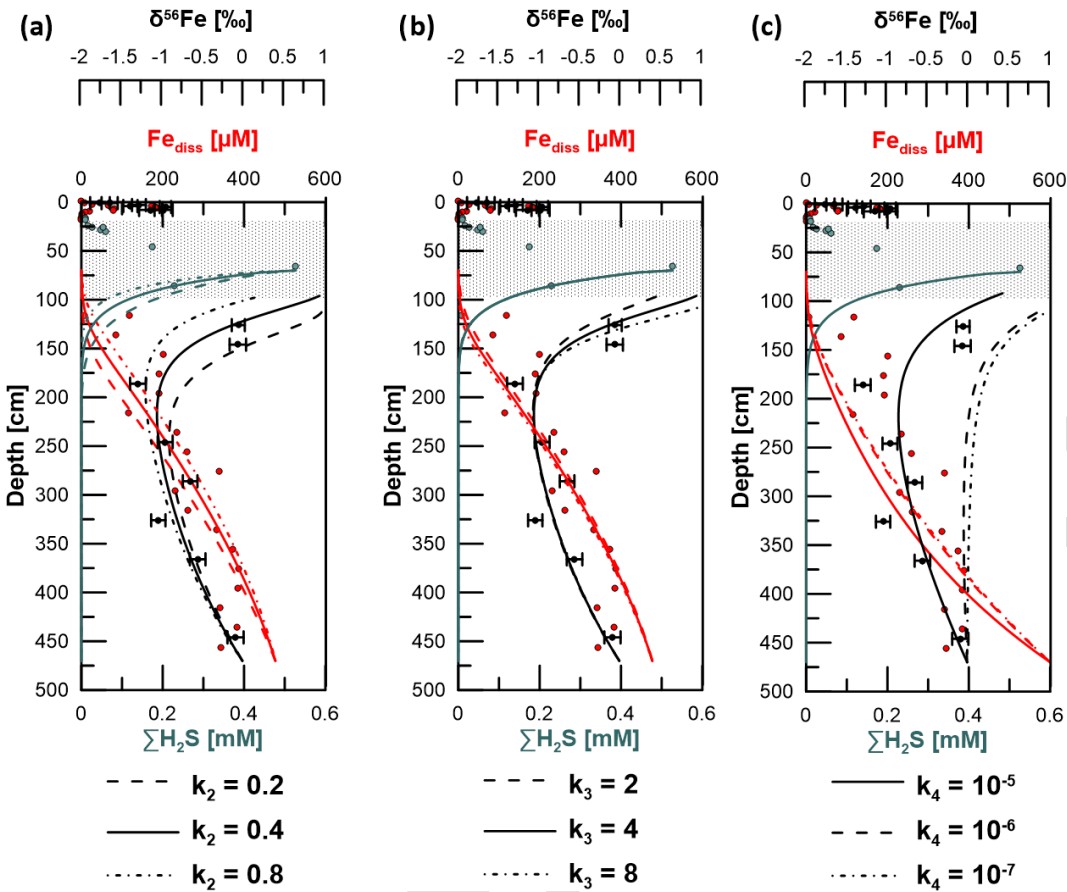

**Figure 8.** Sensitivity test by application of different reaction rate constants ($k$) for **(a)** Reaction (R2) (reaction of free hydrogen sulfide with solid Fe-oxides), **(b)** Reaction (R3) (iron sulfide formation by reaction of $Fe^{2+}$ with $HS^-$), and **(c)** Reaction $R_4$ [TS3] (adsorption of $Fe^{2+}$). Solid lines represent the best fit to the measured data and were thus used in the model to determine kinetic fractionation factors (see Table S2). Uncertainty bars are 2 SD.

over time, intervals with high Fe/Al ratios and Fe contents (e.g., 230–300 cm) reflect enrichments of Fe due to diagenetic Fe mineral formation. In contrast, the intervals of lower Fe/Al and $Fe_{total}$ contents indicate loss of Fe through early diagenetic Fe reduction and subsequent diffusion. However, Fe-oxide dissolution and secondary mineral formation might have been blurred in the record if they happened while the sediment was still in the zone affected by bioturbation. The release of $Fe_{diss}$ into the pore water, in particular at $\sim 5$ and 400 cm depth, and the presence of Fe monosulfides and pyrite in the whole sediment column generally reflect the fact that Fe phases at site HE443/10 undergo considerable early diagenetic transformation. Iron sulfides are indicative of the reaction of solid Fe(III) or $Fe^{2+}$ with hydrogen sulfide, which is released during organoclastic or methane-mediated sulfate reduction (e.g., Poulton et al., 2004; Jørgensen and Kasten, 2006; Riedinger et al., 2017). The peak in $H_2S$ indicates that sulfate-mediated AOM takes place at $\sim 70$ cm. There is a higher diffusive flux $J_{sed}$ of $HS^-$ (ca. $-13\,\mathrm{mmol\,m^{-2}\,yr^{-1}}$) compared to $Fe_{diss}$ ($0.60\,\mathrm{mmol\,m^{-2}\,yr^{-1}}$) towards the sul-

fidization front at 100 cm (Fig. 2). Consequently, there is not only precipitation as FeS, but also formation of pyrite from these monosulfides and Fe-oxides: the removal of $HS^-$ from pore water by sulfide formation or reoxidation exceeds the removal of $Fe_{diss}$.

The most reactive Fe-(oxyhydr)oxides with respect to $H_2S$ are hydrous ferric oxides, ferrihydrite, and lepidocrocite, followed by goethite, magnetite, and hematite (Findlay et al., 2020; Michaud et al., 2020; Poulton et al., 2004). The comparably high $Fe_{hyam}$ contents at site HE443/10 indicate that the amount of $H_2S$ has never been high enough or the time the sediment was subjected to sulfidic conditions has never been long enough to lead to a complete transformation of the highly reactive Fe-oxide pool into iron monosulfides or pyrite. The high $Fe_{hyam}$ contents are at least partly attributed to lepidocrocite, which has previously been detected by Mössbauer spectroscopy in methanic sediments of the HMA (Oni et al., 2015a). It needs to be noted that recent incubation studies showed that the term "reactive" Fe-oxides as we use it here based on chemical extraction is not neces-

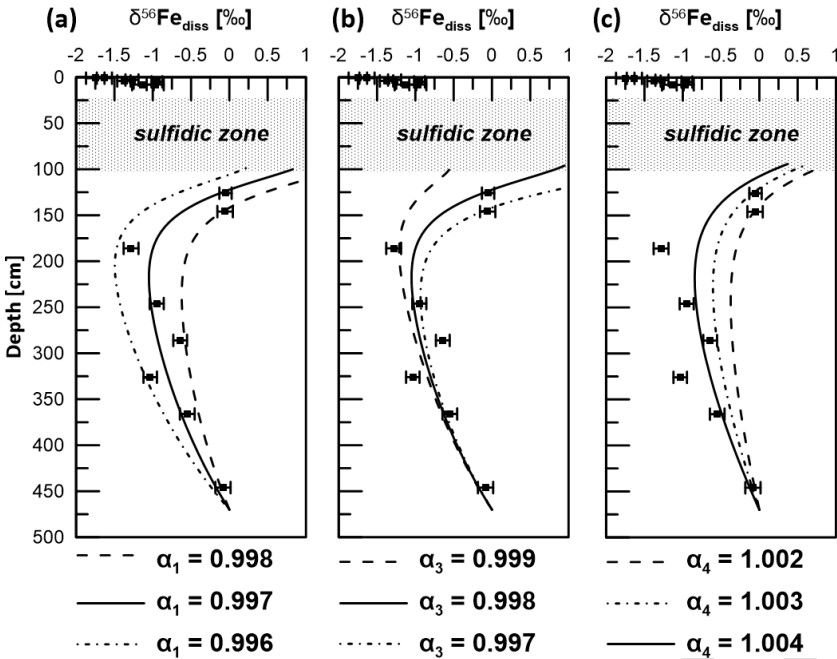

**Figure 9.** $\delta^{56}Fe_{diss}$ profiles derived by applying a transport-reaction model and different kinetic fractionation factors for **(a)** microbial iron reduction ($\alpha_1$), **(b)** pyrite formation via $Fe^{2+}$ and $HS^-$ counter-diffusion ($\alpha_3$), and **(c)** adsorption of $Fe^{2+}$ ($\alpha_4$). Solid lines represent the best fit to measured data (filled circles). Uncertainty bars are 2 SD.

sarily identical to the fraction that is "biologically available" (Aromokeye et al., 2020; Wunder et al., 2024).

There is an enrichment in Fe sulfides (mostly CRS) within the current sulfidic zone and in particular at or close to the up-[5]per sulfurization front at 10 cm depth, where $Fe_{diss}$ and $H_2S$ react to FeS, which is subsequently transformed into pyrite (Fig. 4). This enrichment shows that despite the strong re-working of the sediment, the sulfidic zone must have been fixed to this interval for some years, which is consistent [10]with the low sedimentation rates of $< 3\,mm\,yr^{-1}$ during the past $\sim 700$ years. The abovementioned $Fe_{total}$ enrichment be-tween 230 and 300 cm is not related to a CRS maximum, so it does not represent a paleo-SMT. It is rather attributed to the $Fe_{aca}$ pool, potentially indicating a diagenetic forma-[15]tion of Fe-carbonates (siderite) or AVS. (At site HE443/10, siderite is oversaturated below 100 cm depth; see Fig. S1 in the Supplement.) The Fe enrichment is also, to a lesser ex-tent, reflected by $Fe_{hyam}$ and $Fe_{di-ct}$, which indicates that this interval either has been "recharged" a lot with diagenetic Fe-[20]oxides by oxidation of the upwardly diffusing $Fe^{2+}$ when this interval was close to the surface or was buried more rapidly than sediment above and below so that Fe-oxides were not affected as much by pyritization. Since the interval does not show stronger signs of bioturbation than the rest of the core, [25]which could favor a strong reoxidation of $Fe^{2+}$ and FeS, and since sedimentation rates were high when this interval ac-cumulated, we consider the faster burial through the sulfidic zone to be the more plausible explanation for the Fe enrich-ment observed between 230 and 300 cm.

The extraction with sodium acetate targets Fe associ-[30]ated with carbonates (Poulton and Canfield, 2005; Tessier et al., 1979) but is known to also partly dissolve, for exam-ple, monosulfides (AVS) (Cornwell and Morse, 1987; Poul-ton and Canfield, 2005) and surface-reduced Fe(II) (Crosby et al., 2005, 2007; Henkel et al., 2016). As we extracted AVS [35]separately, we could attribute 6 % to 26 % of the $Fe_{aca}$ pool in samples from site HE443/10 to AVS. Particularly high con-tributions of AVS to $Fe_{aca}$ are found at 116 cm, just below the current depth of the lower sulfidization front; at 236 cm (depth of the abovementioned Fe enrichment); and at 396 cm. [40]The presence of AVS below the sulfidic zone is consistent with the findings of Riedinger et al. (2017) that show that metastable authigenic iron monosulfides can survive burial into deeper sediments in highly dynamic depositional sys-tems, where high sedimentation rates and high Fe-oxide con-[45]tents limit the exposure to free $HS^-$, which is restricted to a narrow zone around the SMT.

Diagenetic processes and reaction fronts in marine sedi-ments are ultimately determined by the quality and amount of accumulated TOC (e.g., Rullkötter 2006). In order to as-[50]sess whether reaction fronts might have shifted upwards or downwards in the past, in particular as a consequence of the abovementioned decrease in sedimentation rates and a poten-tial change in organic matter accumulation, we determined TOC contents. The preserved TOC contents at site HE443/10 [55]are all below 1.2 wt %. Considering some uncertainty as our study site is 5 km southwest of the core dated by Hebbeln et al. (2003), the shift in sedimentation rates should relate

to a depth between 1 and 2 m at HE433/10. Even though TOC values scatter between 0.4 wt % and 1.2 wt %, there is no clear shift that would hint at this drastic change in depositional conditions. We conclude that the overall composition of material that was deposited in the HMA before and after $\sim 700$ BP was similar and that the change was largely limited to the amount of material that was supplied. This is in line with previous observations by Oni et al. (2015b), who suggested similar sources of sediments and organic matter at and below the depth of the SMT based on low variation in $\delta^{13}$C of TOC.

## 4.2 Iron isotope fractionation

As in other marine environments, where in situ Fe reduction in shallow sediments has been observed (e.g., Henkel et al., 2016, 2018; Homoky et al., 2009; Severmann et al., 2006), there is, above the sulfidic zone, an overall downcore trend of $\delta^{56}$Fe$_{\mathrm{diss}}$ towards heavier values ($-1.75$‰ at 1 cm vs. $-1$‰ at 8 cm). This trend is related to (1) preferential removal of light Fe isotopes from the reducible ferric Fe pool during burial and ongoing MIR as well as to (2) preferential removal of light Fe isotopes during interactions with hydrogen sulfide at the sulfidization front (Severmann et al., 2006). The availability of $^{54}$Fe(III) is highest close to the oxic–anoxic boundary, which is reflected by the most negative $\delta^{56}$Fe$_{\mathrm{diss}}$ in the pore water. The processes above the sulfidic zone of HMA sediments were described earlier by Henkel et al. (2016). The easily reducible Fe(III) pool (Fe$_{\mathrm{hyam}}$) contains Fe-oxides that formed from isotopically light Fe$_{\mathrm{diss}}$ and is therefore also isotopically light compared to less reactive Fe-oxide pools. This has already been shown for shallow HMA sediments (Henkel et al., 2016) but can also be observed for the deeper sediments investigated here ($\delta^{56}$Fe$_{\mathrm{hyam}}$ $-0.19 \pm 0.16$‰ (2 SD) compared to $\delta^{56}$Fe$_{\mathrm{di-ct}}$ $0.19 \pm 0.08$‰ and $\delta^{56}$Fe$_{\mathrm{oxa}}$ $-0.09 \pm 0.10$‰).

Dissolved Fe concentrations measured right below the sulfidic zone, at $\sim 130$ cm, are $\sim 100$ µM. Respective $\delta^{56}$Fe$_{\mathrm{diss}}$ values are $\sim 0$‰ (compared to $-1.28$‰ at $\sim 190$ cm, from where Fe is diffusing upwards). At first glance, this is in accordance with observations by Severmann et al. (2006) in sediments from Monterey Bay and Santa Barbara Basin and in sediments from Lake Kinneret (Sivan et al., 2011), where the formation of amorphous Fe sulfides drives $\delta^{56}$Fe$_{\mathrm{diss}}$ towards positive values by preferential removal of $^{54}$Fe from pore water. Experimental studies show that the $\delta^{56}$Fe composition of FeS can be highly variable and depends on proportions of isotope exchange between particle and Fe$_{\mathrm{diss}}$ during aging of iron monosulfides (Guilbaud et al., 2010). Nevertheless, there is general agreement that kinetic isotope fractionation, which dominates in natural sediments, leads to an isotopically light composition of amorphous Fe monosulfides compared to Fe$_{\mathrm{diss}}$ with $\Delta^{56}$Fe$_{\mathrm{FeS-Fe(II)diss}} = -0.85 \pm 0.30$‰ ($\alpha_{\mathrm{FeS-Fe(II)diss}} = 0.999$) (Butler et al., 2005; Roy et al., 2012).

The $^{54}$Fe is further preferentially incorporated into pyrite (with FeS as precursor), as was shown by Guilbaud et al. (2011) for abiotic pyrite formation. Here, $\Delta^{56}$Fe$_{\mathrm{FeS_2-FeS}}$ is $-1.70$‰ to $-3.0$‰ ($\alpha_{\mathrm{FeS_2-FeS}} = 0.998$ to $0.997$), so the combined fractionation factor $\alpha_{\mathrm{FeS_2-Fe(II)diss}}$ (that can be compared to our $\alpha_3 = 0.998$, Fig. 9b) is $0.996$ to $0.997$. It needs to be considered that Fe sulfides age and exchange Fe isotopes with their surroundings (equilibrium fractionation). This is a process which takes place continuously in marine sediments. It might not be dominant, especially not at reaction fronts, but it causes a continuous equilibration of the Fe isotopic composition of different pools, also below the sulfidic interval.

At a second glance, however, it becomes apparent that there is a mismatch between the modeled Fe$^{2+}$ and HS$^-$ profile for this process at site HE443/10 and the respective measured data (Fig. 8b). From its source (AOM at $\sim 70$ cm depth) HS$^-$ diffuses downwards and is used up already at a depth of $\sim 120$ cm, where Fe$_{\mathrm{diss}}$ is already at 118 µM. Based on the model output, HS$^-$ would diffuse down to a depth of almost 150 cm and Fe$_{\mathrm{diss}}$ concentrations at $\sim 130$ cm would be close to zero. Furthermore, although we want to be cautious not to over-interpret a single data point, an Fe$_{\mathrm{diss}}$ sink is indicated by our measured pore-water profile at $\sim 135$ cm (loss of $\sim 50$ % of the upwardly diffusing Fe). We conclude that siderite precipitation might occur at this specific interval. As for the precipitation of FeS, siderite formation would preferentially transfer $^{54}$Fe into the solid phase ($\Delta^{56}$Fe$_{\mathrm{siderite-Fe(II)diss}} = -0.48 \pm 0.22$‰; Wiesli et al., 2004). Unfortunately, the Fe$_{\mathrm{aca}}$ contents vary overall too much to resolve where there is or has been a particular interval affected by siderite formation, and $\delta^{56}$Fe$_{\mathrm{aca}}$ in this potentially affected interval is similar to the $\delta^{56}$Fe$_{\mathrm{aca}}$ directly above and below (Fig. 10). DIC concentrations are too high ($> 50$ mM) to reflect a sink on the order of less than 2 mmol m$^{-2}$ yr$^{-1}$ as calculated from the loss of Fe$_{\mathrm{diss}}$ at 135 cm.

A Keeling plot (Fig. 6b) was used to determine the porewater Fe isotope endmember for the observed deep Fe release. Here, we only used data from below those depths at which $\delta^{56}$Fe$_{\mathrm{diss}}$ is mainly controlled by the reaction with H$_2$S, i.e., between 450 and 150 cm, where there is a rather linear $\delta^{56}$Fe$_{\mathrm{diss}}$ trend (see Fig. S2 in the Supplement for reaction rates of Reactions R1–R3). Although there is a linear trend between $1/[\mathrm{Fe}_{\mathrm{diss}}]$ and $\delta^{56}$Fe$_{\mathrm{diss}}$ ($R^2 = 0.43$), the correlation is not statistically significant ($p$ value 0.161), which is partly due to the low number of data points. The 95 % confidence interval covers a wide range between $-1.4$‰ and $+3.0$‰ for the inferred Fe source (the intercept with the $y$ axis); it is not possible to determine the endmember without a large error. However, the Fe liberation at depth is most likely not causing a preferential release of $^{54}$Fe. The lowermost $\delta^{56}$Fe$_{\mathrm{diss}}$ value is $-0.08 \pm 0.10$‰ and has thus an isotopic composition which is similar (within uncertainty) to the

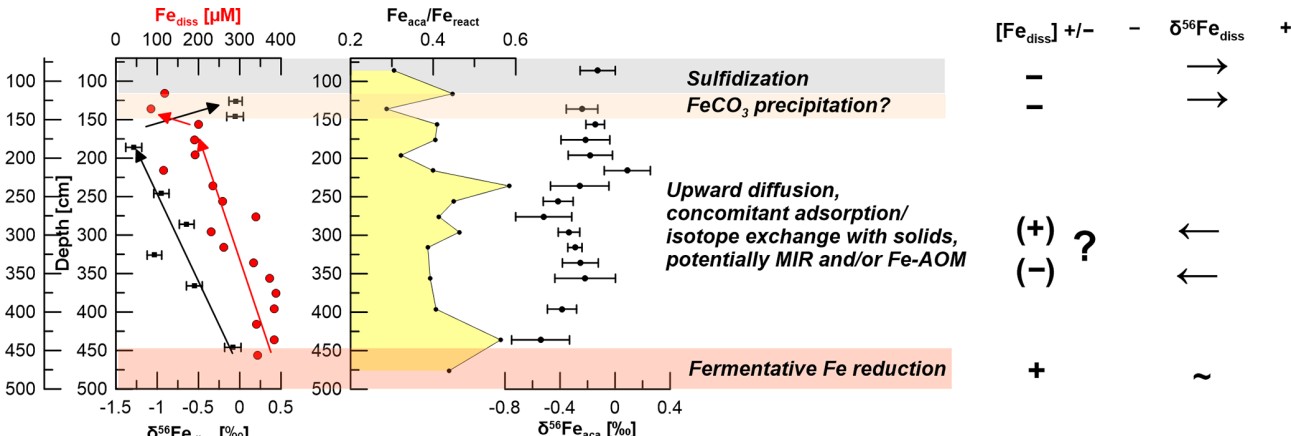

**Figure 10.** Interpretation of pore-water and stable Fe isotopic data for site HE443/10: the deep Fe release does not coincide with significant Fe fractionation and might be explained by fermentative processes. The red arrows show the overall trend of $Fe_{diss}$. The black arrows show the respective trends of $\delta^{56}Fe_{diss}$ values. The upward trend towards lighter isotopic values (between 450 and $\sim 175$ cm) might be explained by either low rates of microbial iron reduction (MIR), preferential removal of $^{56}Fe$ by adsorption, or both. Additionally, Aromokeye et al. (2020) showed low rates of Fe-AOM in incubations of methanic HMA sediments. This process might also affect $\delta^{56}Fe_{diss}$. The shift towards positive $\delta^{56}Fe_{diss}$ values at $\sim 150$ cm likely indicates siderite precipitation.

composition of $Fe_{hyam}$ and $Fe_{oxa}$ and is only slightly lighter than $Fe_{dith}$ (Fig. 5).

The $\delta^{56}Fe_{diss}$ value at $\sim 190$ cm is $-1.28 \pm 0.10\,‰$ (2 SD), so while diffusing upwards, the $Fe_{diss}$ either (1) loses heavy isotopes or (2) is affected by an additional process providing light isotopes. Removal of $^{56}Fe$ from pore water happens by preferential adsorption onto (Fe-oxide) particles, as has recently been shown by Köster et al. (2023). Over time, this Rayleigh distillation process progressively lowers $\delta^{56}Fe_{diss}$ values. Köster et al. (2023) used the fractionation factors $\Delta^{56}Fe_{Fe(II)_{sorb}-Fe(II)_{diss}} = 0.87\,‰$ and $1.24\,‰$ for adsorption of Fe(II) onto goethite surfaces (Beard et al., 2010; Crosby et al., 2007) to calculate the proportion of $Fe_{diss}$ that would need to be "removed" (adsorbed) in order to obtain the extremely negative $\delta^{56}Fe_{diss}$ values that they measured in deep, lithified sediments from the Nankai Trough. Using the same approach here with $\delta^{56}Fe_{diss} = 0\,‰$ for the deep Fe source, between 65 % and 75 % of the $Fe_{diss}$ pool would need to be adsorbed to achieve a value of $-1.28\,‰$ at 190 cm. The $Fe_{diss}$ concentration at $\sim 190$ cm is about 50 % of $[Fe_{diss}]$ at 400 cm (Fig. 2). It is likely that adsorption (and also electron and atom exchange with different Fe minerals) takes place, which means that the concentration profile of $Fe_{diss}$ in the methanic zone is not solely controlled by $Fe_{diss}$ release at depth and Fe–S reactions as a sink. Therefore, by implementing a reaction for adsorption based on Wang and Van Capellen (1996; see Table S1) and the rate expression $R_4 = -k_4[Fe^{2+}]$ with $k_4 = 10^{-5}$, $10^{-6}$, or $10^{-7}$ in our model, we tested if adsorption of $Fe^{2+}$ plays a significant role (apart from being already included into the DIR fractionation factor $\alpha_1$ as described in Sect. 2.5). $k_4$ includes the number of unoccupied surface sites, which is unknown. The test results do not produce a

good fit between the modeled and measured $Fe_{diss}$ profiles (Fig. 8c). Including $\alpha_4 = 1.002$, 1.003, or 1.004 for the respective preferential removal of the heavy isotope (in combination with $k_4 = 10^{-5}$) also did not lead to a good reproduction of the observed distinct shift in $\delta^{56}Fe_{diss}$ to $-1.28\,‰$ at $\sim 190$ cm (Fig. 9). Furthermore, the adsorbed (heavy) Fe should be transferred to the $Fe_{aca}$ pool, but $\delta^{56}Fe_{aca}$ instead shows a trend towards more negative $\delta^{56}Fe_{aca}$ values with depth (Fig. 10). We conclude that adsorption in the methanic sediments at site H433/10 is not the process dominating Fe removal from pore water and the trend towards lighter $\delta^{56}Fe_{diss}$ values between 450 and $\sim 190$ cm.

Our PHREEQC calculations also indicate vivianite saturation in the methanic sediments. Vivianite is a ferrous iron phosphate mineral, $Fe_3(PO_4)_2 \cdot 8H_2O$, which is known to be stable under anoxic sedimentary conditions, but its identification and quantification are difficult. Rothe et al. (2014) found that supersaturated pore water alone does not reliably predict vivianite formation. Furthermore, there are no data available yet concerning the fractionation of iron during vivianite formation. Vuillemin et al. (2020) determined negative $\delta^{56}Fe$ values in vivianite crystals in more than 20 m deep sediments from Lake Towuti, Indonesia, but the dataset does not include pore-water data. We conclude that we do not have enough data to assess the role of vivianite precipitation at our study site. However, vivianite formation would act as a phosphorous sink, and the $PO_4$ profile indicates a sink at $\sim 300$ cm. A slight shift in $\delta^{56}Fe_{diss}$ values towards negative values is recorded slightly below at 325 cm (Fig. 10). Therefore, one could speculate that vivianite precipitation happens and preferentially incorporates $^{54}Fe$.

For the model, we tested MIR as the factor controlling the trend of $\delta^{56}\text{Fe}_{\text{diss}}$ values towards $-1.28\,‰$ at between 450 and $\sim 190\,\text{cm}$ depth. We did so because MIR was stimulated in incubations of methanic sediments from this site when easily reducible Fe-oxides (lepidocrosite) and benzoate were provided (Aromokeye et al., 2021). Therefore, Fe reducing bacteria are present in these sediments, and Oni et al. (2015a) have also shown the presence of lepidocrosite. We also see a concurrent net Mn release in methanic sediments between 150 and 200 cm but only at low levels, potentially because Mn solubility in non-sulfidic sediments is generally controlled by rhodochrosite or other mixed Mn/Ca-carbonates (e.g., Gingele and Kasten, 1994). The net Mn release hints at microbial metal reduction (overlap of MnO$_2$ reduction and MIR). In natural methanic sediments, organoclastic DIR with fermentation intermediates such as acetate is generally assumed to not play an important role because the related bacteria need acetate or other intermediates of fermentation products, which are typically not available at these depths. However, a recent study at a comparable site in the Bay of Aarhus (Kattegat) showed higher acetate concentrations below the SMT than above (Glombitza et al., 2019). Furthermore, through microbiological enrichment experiments, Aromokeye et al. (2021) demonstrated the potential of Fe liberation in the deeper sediments of the HMA being related to the activity of fermenting bacteria, and those are known to produce acetate (Lovley and Phillips, 1986). The OM in methanic sediments was previously characterized as recalcitrant O-rich aromatic and highly unsaturated compounds of terrestrial origin (Oni et al., 2015b), but the authors' methods could not resolve the distribution of low-molecular-weight compounds such as short-chain fatty acids (including acetate). Microbes specialized in recalcitrant aromatic OM degradation often require initial fermentation of the OM to fermentation intermediates (e.g., volatile fatty acids or reducing equivalents, i.e., H$_2$ and acetate) that can be accessed by dissimilatory iron-reducing organisms. In the methanic zone, fermentation intermediates such as acetate and H$_2$ are likely electron donors for methanogenesis, whereas in surface sediments, organic fermentation products are often the electron donors for anaerobically respiring microorganisms with available electron acceptors (sulfate, iron oxides) (e.g., Beulig et al., 2018; Jørgensen, 2006; Whiticar, 1999; Yin et al., 2024). In our model, the applied MIR rate that produces a good fit to the measured $\delta^{56}\text{Fe}_{\text{diss}}$ data is very low (0.00011 mM yr$^{-1}$, Fig. S2) and does not explain Fe$_{\text{diss}}$ concentrations of almost 400 µM at depth. When MIR takes place, the reactive ferric pool typically decreases with depth and becomes enriched in $^{56}\text{Fe}$. The rate is, however, too low to reflect this in the solid-phase data. The reactive Fe pool does not decrease with depth, and downcore isotopic trends are also absent for Fe$_{\text{hyam}}$, Fe$_{\text{dith}}$, and Fe$_{\text{oxa}}$ (Fig. 4). We used a simple calculation to test how strongly the isotopic composition of the ferric Fe pool (here Fe$_{\text{hyam}}$) would change just by MIR at the rate applied in our model. The $\delta^{56}\text{Fe}_{\text{hyam}}$

at depth $L$ can be calculated according to the mass balance equation:

$$\delta^{56}\text{Fe}_{\text{hyam}}^0 \times \text{Fe}_{\text{hyam}}^0 = \delta^{56}\text{Fe}_{\text{hyam}}^L \times \text{Fe}_{\text{hyam}}^L + (1-\alpha_1)\Delta\text{Fe}, \quad (4)$$

where $\text{Fe}_{\text{hyam}}^0$ and $\text{Fe}_{\text{hyam}}^L$ are the weight percent of Fe$_{\text{hyam}}$ at the sediment surface and depth $L$ (m), respectively, with its Fe isotope value $\delta^{56}\text{Fe}_{\text{hyam}}^0$ and $\delta^{56}\text{Fe}_{\text{hyam}}^L$. $\Delta\text{Fe}$ is the amount of Fe that was lost due to Fe$_{\text{diss}}$ release by MIR; i.e., $\Delta\text{Fe} = \text{Fe}_{\text{hyam}}^0 - \text{Fe}_{\text{hyam}}^L$ With a sedimentation rate of $\omega = 0.0016\,\text{m yr}^{-1}$ and an MIR rate of $R_1 = 0.00011\,\text{mM yr}^{-1}$, $\Delta\text{Fe}$ is calculated as follows:

$$\Delta\text{Fe} = \frac{4R_1 L M_{\text{Fe}}\varphi}{\rho_s(1-\varphi)} \times 100, \quad (5)$$

where $M_{\text{Fe}}$ is the molecular weight of iron. According to this, the Fe$_{\text{hyam}}$ pool would lose only 0.014 wt % (0.14 mg g$^{-1}$) between the sediment surface and 5 m depth. The isotopic difference in Fe$_{\text{hyam}}$ between the surface and 5 m depth would be 0.1‰, which is in the range of our analytical uncertainty (2 SD).

We note that Aromokeye et al. (2020) also demonstrated that Fe-AOM occurs at low rates, in particular right below the sulfidic zone. This process that releases 8 mol of Fe$_{\text{diss}}$ for each mole of CH$_4$ might potentially also play a role. But as there are no literature data on the respective Fe fractionation, our study cannot resolve whether it is solely MIR or MIR and Fe-AOM occurring at low rates (Fig. 10).

We observe that high Fe$_{\text{dith}}$ contents are related to slightly higher $\delta^{56}\text{Fe}_{\text{dith}}$ values (Fig. 6a), a situation that is counterintuitive assuming that $^{54}\text{Fe}$ would be preferentially lost when the Fe$_{\text{dith}}$ pool is reduced by microbes, so lower contents should coincide with a more positive $\delta^{56}\text{Fe}_{\text{dith}}$ value. The Miller–Tans plot should therefore not be interpreted as reflecting downcore trends: Fe$_{\text{dith}}$ (as well as Fe$_{\text{aca}}$, Fe$_{\text{hyam}}$, and Fe$_{\text{oxa}}$) maxima are recorded within the methanic interval (Fig. 4). The plot demonstrates that the isotopic differences between sequentially extracted Fe pools are largest (but still small) where Fe contents are highest – a circumstance that is also unexpected since the isotopic fractionation in the (remaining) substrate should be expressed more strongly if the ferric Fe pool is small. This is possibly an effect of non-steady-state conditions in the past. Overall, the Fe$_{\text{oxa}}$ pool is not affected by Fe isotope fractionation, but all other extracted Fe(III) pools are. The pools that are known to contain sorbed Fe(II) and typical secondary minerals (Fe$_{\text{aca}}$ and Fe$_{\text{hyam}}$) form from (isotopically light) Fe$_{\text{diss}}$ and are therefore characterized by overall negative $\delta^{56}\text{Fe}$ values and, as expected, lighter composition at higher contents. Only the $\delta^{56}\text{Fe}_{\text{dith}}$ pool shows a slight shift towards positive values at higher contents, which we cannot entirely resolve here. Considering that the iron released at depth has an isotopic composition close to 0‰, adsorbed iron deriving from upward diffusion would potentially have a positive $\delta^{56}\text{Fe}$ signature. If part of the adsorbed (heavy) iron is then exchanged with

the reactive Fe-oxide surface (Crosby et al., 2007) and might subsequently even migrate deeper into the iron oxide crystal (Larese-Casanova et al., 2023), it could cause an alteration of Fe-oxide isotope signatures towards positive values without reducing the mineral. It might also be speculated that adsorption and the related electron and atom exchange are more prevalent at depths that have a high Fe-oxide ($Fe_{dith}$) content, but this interpretation remains very speculative, in particular because our model does not indicate adsorption to be a dominant Fe sink.

## 4.3 Deep iron release

Deep iron reduction can occur purely abiotically via oxidation of reduced sulfur compounds (e.g., Holmkvist et al., 2011; Riedinger et al., 2010). However, Fe reduction due to this so-called cryptic S cycling fails as an explanation for the buildup of $Fe_{diss}$ far below the sulfidic zone, as has also been concluded by Riedinger et al. (2014) for Fe-rich continental-margin sediments off Argentina, by Egger et al. (2014) for Bothnian Sea sediments, and by Oni et al. (2015a) for the methanic sediments of the HMA.

Respiratory methanogenic iron reduction (e.g., Sivan et al., 2016; Eliani-Russak et al., 2023; Gupta et al., 2024, and references therein) might be a possible explanation for deep Fe release in methanic sediments. However, as the process is respiratory, we assume it would lead to Fe isotope fractionation similar to MIR in shallow sediments. Furthermore, in order to perform respiratory Fe(III) reduction, methanogens would need to oxidize $CH_4$ or an organic substrate, e.g., acetate or methyl compounds. It seems unlikely that methane oxidation would support growth coupled to Fe(III) reduction (see Chadwick et al., 2024). Gupta et al. (2024) stated for example that "even though we and others have shown that methanogens like *M. acetivorans* are metabolically active and can conserve energy by iron respiration . . . , it is still not known whether methanogens can couple iron reduction to growth in addition to energy conservation." The correlation of $Fe_{diss}$ concentrations with JS1 bacteria, methanogens, and *Methanohalobium*/ANME-3-related archaea at our study site suggested that the deep Fe reduction is coupled to the activity of these microbes (Oni et al., 2015a). An overlap of hydrogenotrophic $CH_4$ production and Fe-AOM has also been proposed based on the isotopic composition of $CH_4$ in sediments from the Baltic Sea (Egger et al., 2014, 2017). Several incubation experiments have demonstrated that the addition of reducible Fe-oxides can stimulate Fe-AOM in natural sediments characterized by low/absent sulfate concentrations (Aromokeye et al., 2020; Beal et al., 2009; Egger et al., 2014; Segarra et al., 2013; Sivan et al., 2011). Aromokeye et al. (2020, 2021), however, found that in incubations of HMA sediment, Fe release occurred not entirely through Fe-AOM but was largely unrelated to methane oxidation and seemed to be instead linked to the fermentation of complex organic matter – a pro-

cess that can be stimulated by crystalline Fe-oxides because (1) fermenters reduce Fe(III) as an outlet for electrons primarily to overcome the thermodynamic barriers caused by high concentrations of newly produced fermentation intermediates, thus enabling continued OM degradation (fermentative iron reduction; Hopkins et al., 1995; Lovley, 1991), and (2) the conductive character of the Fe-oxides facilitates interspecies electron transfer from fermenting bacteria towards methanogens (Kato et al., 2012; Lovley and Holmes, 2022).

Building on all these previous studies, our iron isotopic data further hint at a deep Fe release that is not linked to DIR or another process in which microbes would preferentially use isotopically light Fe-oxides. Fermenting bacteria typically require a syntrophic partner such as an $H_2$-utilizing bacterium (Hopkins et al., 1995) or a methanogen (e.g., Kato et al., 2012). The syntrophic partner consumes fermentation intermediates as a primary pathway for electron release and for thermodynamic feasibility of OM degradation. In the absence of a syntrophic partner, fermenting bacteria have been shown to be capable of electron transfer to iron oxides to further promote OM degradation. The iron oxides may be reduced fortuitously in the process as a final sink for the electrons or serve as a conduit, further transferring the electrons to an available syntroph, e.g., a methanogen (Aromokeye et al., 2021). The fermenting bacteria that transfer electrons to crystalline Fe-oxides do not directly profit from Fe(III) reduction beyond the removal of thermodynamic limitations brought about by accumulation of fermentation intermediates. In other words, the fermenters use the conductive Fe-oxides to transfer electrons and to be able to continue with the fermentation of particularly aromatic OM. The transfer of electrons via conductive Fe-oxides speeds up the degradation of aromatic compounds and is metabolically and mechanistically beneficial to both partner microbes (e.g., Jiang et al., 2013; Kato et al., 2012; Cruz Viggi, 2014; Zhuang et al., 2015). Meanwhile some doubt is building up that those electrons are really conducted in an electronic fashion without reduction and reoxidation of iron occurring. This is summarized in a review article by Xu et al. (2019). Figure 11 summarizes how fermenting bacteria, MIR-performing bacteria, and methanogens are known to interact and how the deep iron release as observed in the sediments of the HMA could be explained. It is known that in subseafloor sediments, there is a cooperative exchange of electrons and hydrogen in microbial communities and that this is also happening during fermentation (Shah et al., 2013). But to the best of our knowledge, there are no studies available that show how fermentative iron reduction takes place mechanistically, i.e., directly or indirectly by the fermenting bacteria or during the interspecies electron transfer. In addition, there are no experimental studies on how fermentative iron reduction fractionates iron isotopes. This is a gap in knowledge that should be addressed by future studies. In any case, the reason for the supposed fermentative Fe reduction happening at depth and not, for example, directly below the sulfidic zone might be selective OM

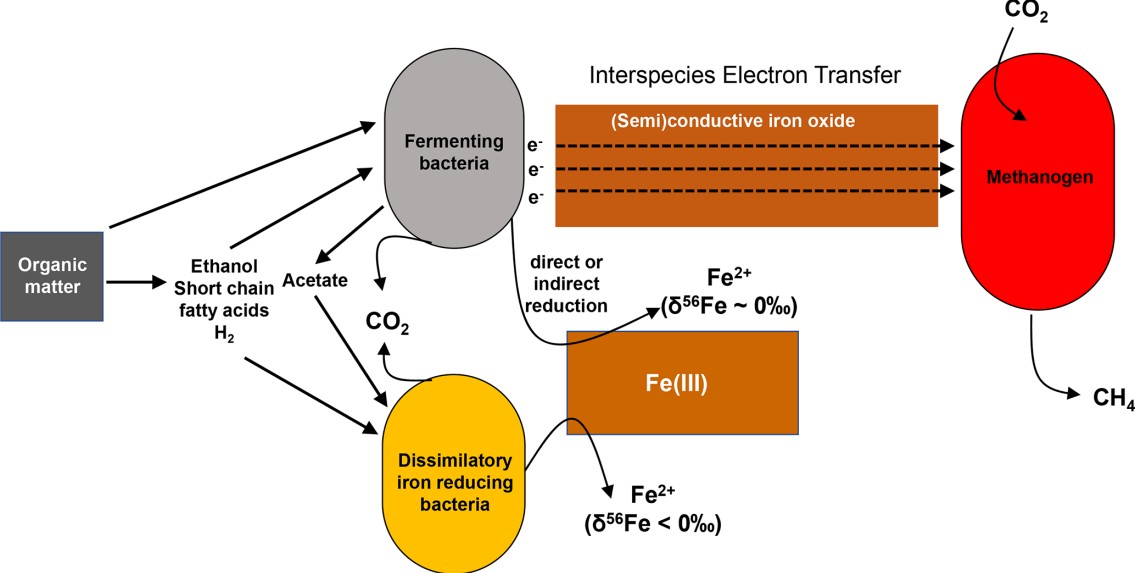

**Figure 11.** Schematic representation of how deep iron reduction in methanic sediments of the HMA could be controlled. The relative contributions of microbial iron reduction (here DIR) and fermentative iron reduction likely depend on the availability of Fe-oxides, the composition of the organic matter, and the abundance of methanogens as "partner organisms" to which fermenters transfer electrons.

degradation with more aromatic or unsaturated compounds in the deep sediment (Gibson and Harwood, 2012; Oni et al., 2015b).

## 4.4 Applicability of iron isotopes to trace iron processes in marine sediments

We demonstrate that the application of iron isotopes in marine sediments provides information that helps in identifying or verifying specific Fe reaction pathways. However, the main difficulty in using iron isotopes in natural systems is that, usually, various processes of Fe liberation and incorporation into solid phases are at play simultaneously. In the deep HMA sediments which contain lepidocrocite (Oni et al., 2015a) as well as crystalline Fe-oxides, different pathways of microbial organic matter oxidation with the involvement of Fe phases are likely to happen simultaneously – namely microbial iron reduction and fermentative processes with electron shuttling. These processes are therefore hard to resolve by iron isotope data alone. Generally, when multiple iron reactions are taking place, the resulting Fe isotope signals in dissolved and solid pools might not reflect or resolve all specific fractionation processes. Furthermore, over time, equilibrium isotope fractionation overprints kinetic isotope fractionation, so the isotopic composition of two pools that are susceptible to atom exchange will change until heavy isotopes are enriched in the pool with "stiffer" bonds (e.g., Wiederhold, 2015). We also note potential challenges when working with sequentially extracted Fe pools. As has been shown previously, these pools are not mineral-specific (e.g., Henkel et al., 2016). If the overall content of a pool is large

compared to the amount of Fe in that pool that was affected by diagenesis, then the respective isotopic differences (e.g., downcore) might still be in the range of the analytical uncertainty. Therefore, depending on the setting, resolving differently reactive Fe phases and analyzing the respective $\delta^{56}$Fe signals might not be specific enough to clearly deduce which processes took place. This study and the comparison to the study of Köster et al. (2023) demonstrate that, unsurprisingly, processes dominating the shape of $\delta^{56}$Fe profiles and records differ depending on depth for two reasons: (1) the microbial community changes in composition and quantity due to depth-dependent availabilities of organic matter and electron acceptors and (2) equilibrium fractionation and processes like adsorption become increasingly important with the age of the studied sediment. In any case, the application of Fe isotopes in marine sediments requires a large set of complementary geochemical and microbiological data to achieve a robust interpretation.

## 5  Summary and conclusions

Here we applied stable iron isotope analyses to pore water and sequentially extracted, differently reactive iron phases and transport-reaction modeling to identify the process responsible for the observed deep iron release in methanic sediments of the Helgoland mud area. The comparison between the isotopic composition of dissolved Fe and the ferric solid substrates reveals that the deep Fe release does not lead to a preferential liberation of $^{54}$Fe as occurs during DIR in shallow sediments. In combination with previous microbial

studies, this isotopic study implies that iron reduction occurs during fermentative iron reduction when electrons are transferred from fermenters to iron oxides. In contrast to DIR, the "choice" of iron isotopes during the reduction seems to be rather coincidental. However, studies on the mechanistic details of fermentative iron reduction (including Fe isotope analyses) are needed to prove our interpretation.

This study provides a concept for how to deal with the complexity of geochemical and in particular Fe isotope data from pore water and sediments in order to test whether specific Fe redox reactions are or are not at play. We conclude that in combination with microbial experiments and geochemical and transport-reaction modeling, basic additional knowledge about Fe reactions can be gained by applying Fe isotope geochemistry. However, data interpretation is still far from being straightforward. This study also demonstrates that robust data interpretation relies on a combination of methods and the involvement of different areas of expertise.

*Data availability.* Data are available in PANGAEA (https://doi.org/10.1594/PANGAEA.893760, Aromokeye et al., 2018a; https://doi.org/10.1594/PANGAEA.893766, Aromokeye et al., 2018b) linked to the publications by Aromokeye et al. (2020) and Henkel et al. (2024a, b) (DOIs: https://doi.org/10.1594/PANGAEA.971531, https://doi.org/10.1594/PANGAEA.971490).

*Supplement.* The supplement related to this article is available online at [the link will be implemented upon publication].

*Author contributions.* SH and SK designed the study. MS, AM, and SAK provided expertise, access to MC ICP-MS instruments, and technical support. BL performed the modeling. SH compiled the geochemical data and wrote the manuscript with contributions from all co-authors. All co-authors were involved in data discussion.

*Competing interests.* The contact author has declared that none of the authors has any competing interests.

ther geographical representation in this paper. While Copernicus Publications makes every effort to include appropriate place names, the final responsibility lies with the authors.

*Acknowledgements.* We thank Ingrid Stimac (AWI), Ingrid Dohrmann (AWI), Jochen Scheld (University of Cologne), and Johann Hollop (at that time a student at the University of Bremen) for their support during sample processing. We particularly thank Clara Sena, who provided the thermodynamic database used in the PHREEQC calculations as well as the sample input files. Gerhard Kuhn (AWI) is thanked for providing TOC and TC data. We thank the two anonymous reviewers for their helpful comments and suggestions.

*Financial support.* This research has been supported by the Deutsche Forschungsgemeinschaft (grant no. 390741603) and the Bundesministerium für Bildung und Forschung (grant no. 03F0874A).

The article processing charges for this open-access publication were covered by the Alfred-Wegener-Institut Helmholtz-Zentrum für Polar- und Meeresforschung.TS4

*Review statement.* This paper was edited by Chiara Borrelli and reviewed by two anonymous referees.

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

**Remarks from the language copy-editor**

CE1 Note that a comma was removed here to enable the use of commas rather than semicolons between list items.

CE2 These can only be changed to commas if the list of chemicals applies to the entire list and not just the aqueous concentrations. Please clarify. If that is the case, then the punctuation can be adjusted and "the" before "aqueous concentrations" should be moved to before "measured DIC". Otherwise it is our standard to use semicolons to separate list items where at least one item contains internal commas.

**Remarks from the typesetter**

TS1 Please note: according to our standards, a skinny space is added in numbers that have 5 or more figures.

TS2 Please give an explanation of why Eqs. (1) and (2) need to be changed. We have to ask the handling editor for approval. Thanks.

TS3 Please confirm "$R_4$"

TS4 Please note that the second sentence is a standard sentence and cannot be changed.

TS5 The DOI has been removed because it does not work.

TS6 Please confirm page range.

TS7 Please provide URL/DOI.