# Peer review of "Stable iron isotope signals indicate a "pseudo-abiotic" process driving deep iron release in methanic sediments"

_EGUsphere, 2024_

## Referee Comment (RC2)

[referee-annotated manuscript omitted]

---

## Author Comment (AC1)

**Responses to Reviewer Comments**

**Manuscript "Stable iron isotope signals indicate a "pseudo-abiotic" process driving deep iron release in methanic sediments" by Henkel et al., https://doi.org/10.5194/egusphere-2024-1942**

RC2:

**General Comments:**

**Overall, this is a very interesting study, presenting a comprehensive dataset and a scientific approach that combines field data with modeling. The illustrations are clear, the manuscript is well-written, and the interpretations, while complex and occasionally probably speculative, remain cautious. This is well-explained throughout the manuscript.**

**This work contributes to advancing our understanding of the iron cycle in methanic sediments. For all these reasons, I recommend the publication of this work.**

**I have two major remarks, in addition to several minor points listed in the PDF as comments:**

- **The authors appear to assume, given the chemical conditions, that dissolved iron is always reduced iron (Fe-II). However, what is referred to as dissolved iron is, in fact, filtered iron (0.1 μm). Was the redox speciation of iron measured in all the pore waters? In the oceanic water column, 'dissolved' iron encompasses a variety of physicochemical species, including small particles (<0.1 μm containing Fe-III) or colloidal complexes (which can also contain Fe-III). The process of 'non-reductive dissolution' of iron – which does not require reductive conditions as it can involve desorption or ligand-promoted dissolution – seems to be an important process for the release of 'dissolved' - in fact filtered - iron at the sediment-seawater interface (Radic et al. 2011, Labatut et al. 2014, Homoky et al. 2021). Could this 'non-reductive dissolution' process play a role in these methanic sediments?**

This is in fact a very important point raised by the reviewer and we will clarify the issue of Fe speciation in the revised version of the manuscript. For the MUC at the investigated station HE443/10 and also for MUCs and GCs at other stations of the same expedition, we measured both, total dissolved Fe (by ICP-OES) and $Fe^{2+}$ by the ferrozine method after Stookey (1970). The profiles match quite well as can be seen here:

[Figure]

**HE443/074-1  SL**

[Figure]

**HE443/010-2  MUC**

So, we are confident that the dFe pool at the sites investigates in the Helgoland Mud Area consists mainly of $Fe^{2+}$ and don't see a reason to include NRD into the discussion. The reason why we don't have $Fe^{2+}$ data for the gravity core samples is that before the addition of ferrozine we usually keep the $Fe^{2+}$ stable by addition of an ascorbic acid solution. For the gravity core samples, however, the dissolved Fe concentrations were too high for the ascorbic acid mixture. Ascorbic acid was not present in excess. This was unfortunate and as consequence we changed the way we treat the samples directly after sampling (direct transfer of sample into cuvettes pre-filled with ferrozine).

Nonreductive iron dissolution (NRD) is a process that is usually brought up to explain heavy Fe isotope signals of dissolved Fe at the very surface or above the surface of the sediment. We are a bit sceptic concerning this, because a heavy dissolved Fe isotope signal can also be produced by the kinetic precipitation of Fe oxides (e.g., Staubwasser et al. 2013) that is without doubt happening at the oxic/anoxic boundary. We are not saying, NRD can't play a role in areas other than the Helgoland Mud Area. However, NRD is in our view for many of the sites for which this process has been proposed to take place, not *needed* in order to explain the pattern of stable iron isotopes in pore water or the deep water when considering that kinetic Fe oxide precipitation leads to a preferential incorporation of light Fe isotopes and only by aging and with equilibrium fractionation, Fe oxides become "more heavy".

- **The validation of the isotopic measurements appears too superficial. Potential artifacts (isotope fractionation, contamination) related to chemistry, preconcentration, purification, and partial dissolution seem not to have been thoroughly investigated. Yields and blanks from the various steps are not consistently reported. Repeatability seems not to have been quantified across the entire protocol (including the chemistry), and the error bars reported in the graphs seem too small. For instance, the repeatability of the instrument for pore waters appears to be 0.26 ‰, which in my view implies that no measurement of pore waters can have an error bar smaller than 0.26 ‰.**

Actually, all these issues (isotope fractionation, process blanks, preconcentration) have been thoroughly investigated, but they have been presented already in the papers Henkel et al. (2016) and Henkel et al. (2018) to which we refer. We will revise the manuscript to point out more clearly, where the respective information can be found. Furthermore, we will revise the figure caption to Fig. 2. In fact, the given uncertainty bars in the figure refer to the 20 cycles of one block of an analysis. So, all the 20 single analyses were done within ~2 minutes during one dip into the respective sample. The repeatability that is given with 0.26‰ in contrast, was determined by measuring the JM standard 15 times (so in fact 15 x 20) during the run of two separate sequences. We also included replicates of pore-water samples (at 1 and 8 cm depth) into the sequences. Those are displayed in Fig. 1 as well.

**see the pdf for other comments**

**Copied in here from the pdf:**

Line 14: "The low $\delta^{56}$Fe values of dissolved iron liberated by microbial iron reduction are characteristic for shallow subsurface sediments and benthic Fe fluxes into the water column."

**Reviewer: I do not agree. In many places benthic Fe fluxes have been characterized by slightly heavy iron isotope signatures.**

This refers again to the potential NRD that has been mentioned before. We will reformulate this sentence a bit and will put more emphasis to the fact that microbial iron reduction in shallow sediments preferentially releases light Fe. We will be more careful with the expression benthic flux *out of the sediment*, because indeed, the isotopic signature of the dissolved Fe pool may be altered in the top few millimeters due to Fe oxide precipitation and/or NRD.

**Line 41: "Iron isotopes, expressed as $\delta^{56}$Fe (‰), are thus considered as a tool for assessing the role of microbial iron reduction (MIR) for the mineralization of organic matter and for tracing benthic iron fluxes into the water column (e.g., Conway and John, 2014; Homoky et al., 2009; Severmann et al. 2006, 2010; Sieber et al., 2021).**

**Reviewer: Better refer to John, S. G., Mendez, J., Moffett, J., & Adkins, J. (2012). The flux of iron and iron isotopes from San Pedro Basin sediments. Geochimica et Cosmochimica Acta, 93, 14–29. https://doi.org/10.1016/j.gca.2012.06.003."**

We will add the paper by John et al. (2012).

**Line 154: "Not sure an article can be accepted with incomplete references."**

This refers to the PANGAEA dataset for which I now included the full reference (incl. doi).

**Line 176: "The measured value was 0.49 ± 0.26‰ (n=15, 2SD) and overlapped within uncertainty with previously published values (0.42 ± 0.05‰, Schoenberg and von Blanckenburg, 2005; 0.46 ± 0.20‰, Walczyk and von Blanckenburg, 2005; 0.35± 0.14‰, Weyer and Schwieters, 2003)."**

**Reviewer: "This means the instrumental precision is 0.26 per mil... Repeatability of the whole procedure, including pre-concentration and purification should have been quantified. 2 duplicate samples are not enough to statistically evaluate repeatability. Trueness of the instrumental analysis has been quantified (no bias), but trueness of the entire procedure is not validated here. Trueness should have been quantified for the whole procedure from pre-concentration to Neptune analysis. Yields and blanks should have been presented."**

We will revise the respective paragraph and include details to the trueness of the entire procedure. The procedure is, however, not new. Details regarding the procedure are for example given in Henkel et al. (2016) and Henkel et al. (2018).

Lines 189-192:

**"Sequential Fe extractions were performed after Poulton and Canfield (2005): ~50 mg of dry sediment were suspended in always 5 ml of a) $MgCl_2$ for adsorbed Fe, b) Na-acetate for Fe-carbonates and surface-reduced Fe(190 II), c) hydroxylamine-HCl for easily reducible Fe-oxides (ferrihydrite, lepidocrocite), d) Na-dithionite/citrate for reducible Fe-oxides (goethite and hematite) and e) ammonium oxalate/oxalic acid for magnetite."**

**Reviewer: "All this partial extractions could be associated with isotope fractionations, leading to procedural artefacts. This should be discussed. On what basis can we conclude that this is not the case?"**

A thorough analysis of these issues is presented in Henkel et al. (2016) to which we refer. But we will revise the text so that readers can easily find the details.

**Line 198-205:**

**"The MC-ICP-MS was equipped with a SSI dual cyclonic spray chamber, a low flow 50 μl PFA nebulizer and a Ni skimmer cone (x-type). Samples were measured using the standard-sample bracketing with certified reference material IRMM-014. All $^{54}Fe$ data were Cr-corrected based on measurements of $^{52}Cr$. In addition, all data were blank-corrected and samples were analysed in random order. The standard JM (see above) was analysed after each block of three samples. Samples bracketed by JMs that did not fall into the target range of 0.42 ± 0.05‰ were repeatedly measured. The repeatability precision resulting from up to 6 replicate sample measurements (not including replicate processing) was better than 0.34‰ (2SD) and on average 0.11‰. The intermediate precision of JMs was 0.44 ± 0.15‰ (n=151, 2SD)."**

**Reviewer: "Not clear why these details are given here but not in the previous section about pore waters."**

The reviewer is right. We will provide more details regarding the pore-water analyses.

Concerning the "on average 0.11‰": **"I do not understand how this number is calculated"**

We will reformulate this. Basically, we wanted to express that we did repeated measurements for several samples that were measured each up to 6 times. In worst case, the repeatability was 0.34‰. For all other samples it was much lower, sometimes 2SD was 0.03‰ for 4 separate analyses of the same sample.

Fig. 2:

[Figure]

**Reviewer: "Error bars are too small. If the JM Fe wire repeatability was 0.26 per mil (2SD), then, it seems to me that, no measurement can be considered more precise than that."**

This issue is addressed in the response to another comment.

**Line 305-306 and Line 321: "Phosphorus concentrations measured by ICP-OES of acidified pore-water aliquots (not shown, but available under xxx)" and "… PANGAEA (xxx)."**

**Reviewer: "Cannot be accepted like that."**

The full reference to the PANGAEA dataset will be included into the revised version.

**Figure caption to Fig. 6: "b) Keeling plot for $\delta^{56}Fe$ values of pore water with 95% confidence interval."**

**Reviewer: "Why only 6 data points. This should be explained in the legend."**

This is true. It was only written in the main text (Line 493-495): "Here, we only used data from below those depths at which $\delta^{56}Fe_{diss}$ is mainly controlled by the reaction with $H_2S$, i.e., between 450 and 150 cm, where there is a rather linear $\delta^{56}Fe_{diss}$ trend…". We will add "We only used data from between 450 and 150 cm, where there is a rather linear $\delta^{56}Fe_{diss}$ trend." to the figure caption.

**Lines 458 and 459: "This trend is related to 1) the progressive removal of $^{54}Fe$ from the reducible ferric Fe pool during burial and ongoing MIR as well as to 2) progressive preferential removal of $^{54}Fe$ during interactions with hydrogen sulfide at the sulfidization front (Severmann et al., 2006)."**

**Reviewer: "preferential removal of light Fe isotopes"**

We will reformulate the sentence according to suggestion.

**Lines 502-503: "The $\delta^{56}Fe_{diss}$ value at ~190 cm is -1.28 ± 0.10‰ (2SD), so while diffusing upwards, the $Fe_{diss}$ either (1) loses $^{56}Fe$ or (2) is affected by an additional process providing $^{54}Fe$."**

**Reviewer: "As noted above, Fe does not just have 2 isotopes. Therefore I believe that the authors should use 'light isotopes' instead of '54' and 'heavy isotopes' instead of '56'"**

Yes, correct! We will also reformulate this sentence.

**Very impressive and very nice work !**

Thank you!

**Citation: https://doi.org/10.5194/egusphere-2024-1942-RC2**

**Stookey (1970)** Ferrozine-A New Spectrophotometric Reagent for Iron, Anal. Chem. 42, 7, 779-781, https://doi.org/10.1021/ac60289a016.
**Henkel et al. (2016)** Determination of the stable iron isotopic composition of sequentially leached iron phases in marine sediments. Chem Geol. 421, 93-102, http://dx.doi.org/10.1016/j.chemgeo.2015.12.003.
**Henkel et al. (2018)** Iron cycling and stable Fe isotope fractionation in Antarctic shelf sediments, King George Island, Geochim. Cosmochim. Acta, 237, 320-338, https://doi.org/10.1016/j.gca.2018.06.042.

---

## Author Comment (AC2)

**Responses to Reviewer Comments**

**Manuscript "Stable iron isotope signals indicate a "pseudo-abiotic" process driving deep iron release in methanic sediments" by Henkel et al., https://doi.org/10.5194/egusphere-2024-1942**

**RC1**

**General comments:**

**In this paper, the authors carried out a thorough analysis of porewater and solid phase chemistry throughout a core of marine sediments, with a focus on patterns of Fe isotope composition. The goal was to use these patterns to explain patterns of Fe biogeochemistry throughout the depth of the sediments. The thoroughness of the differential extractions was really impressive. They were operationally defined (e.g. roughly corresponding to adsorbed Fe(II), poorly crystalline Fe(III), crystalline Fe(III)) but those operational definitions are still meaningful from the standpoint of Fe biogeochemistry. The authors found that patterns of Fe isotope compositions of Fe(II) and Fe(III) phases were inconsistent with those observed to result from dissimilatory/respiratory Fe(III) reduction in lab experiments. The authors indicate that adsorption/atom exchange does not contribute to the observed isotope patterns even though they also indicate that the rates of Fe(III) reduction are likely quite low.**

**I am struggling with the fermenter conclusion a little bit, because it seems like the authors are saying "patterns of Fe isotopes in all of these different pools are inconsistent with dissimilatory/respiratory Fe(III) reduction, so it must be from the fermenters dumping electrons." Additionally, conduction of electrons from fermenters to methanogens would not result in a net reduction of the Fe (the fermenter would reduce it, but the methanogen would oxidize it). Granted, it is likely that an initial reduction would have to occur, because most hypotheses for interspecies electron transfer via Fe involve magnetite or pyrite, but after that initial reduction, no net redox change would occur with the Fe.**

We thank the reviewer for this generally positive feedback. We are aware that the conclusions of this manuscript still remain a bit speculative. The link to the fermenters is actually indicated by the previous microbial study by Aromokeye et al. (2021).

The reviewer wrote "the fermenters would reduce it" (Fe), but this is in fact not easy to prove. Fermenters may use crystalline Fe oxides to conduct electrons towards methanogens. We discuss this as an option that some of the electrons are "redirected" and are used (by fermenters *or* other microbes) for Fe reduction. Kato et al. (2012) and Cruz Viggi et al. (2014). demonstrated the use of Fe oxides as conductors are Meanwhile, some doubt is building up that those electrons are really conducted in an electronic fashion without reduction and reoxidation occurring. This is summarized in the review article by Xu et al. (2019). So, this supports our interpretation and we will revise the text accordingly to point out more clearly that there are many studies that see an enhancement of syntrophic activity in the presence of conductive magnetite or semiconductive iron minerals. But the details on how this mechanistically works on the molecular level and whether it involves the reduction of Fe(III) or not have not been elucidated yet.

The fact that we don't see a significant Fe isotope fractionation at the depth of the deep $Fe^{2+}$ release at our study site indicates that the underlying reduction process is different to what dominates in shallow marine sediments. This absolutely makes sense, because in shallow sediments the electron donor for microbial iron reduction is acetate, which is less abundant in methanic sediments. The latter, in contrast, contain more $CO_2$, and $CO_2$-dependent methane formation is prevalent.

The reviewer states that "after that initial reduction, no net redox change would occur with the Fe" because "the fermenter would reduce it, but the methanogen would oxidize it". As far as we can judge, methanogens have not been conclusively shown to perform iron oxidation. There is a statement by Dinh et al. (2004) that implies this: "Similarly, a newly isolated *Methanobacterium*-like archaeon produced methane with iron ($Fe^0$) faster than do known hydrogen-using methanogens, again suggesting a more direct access to electrons from iron than via hydrogen consumption". But this paper was published 20 years ago and was targeting the oxidation of metallic iron ($Fe^0$); by now we know that DIET (Direct Interspecies Electron Transfer) between fermenting bacteria and methanogens plays an important role. TDinh et al. 2004 is about corrosion of metallic iron. The authors proposed the direct oxidation of metallic iron to $Fe^{2+}$ and electrons taken up by SRB. For methanogens, the picture is less clear based on that study, but it seems to follow the same idea. A more recent study to the same topic is Holmes et al. (2022). However, metallic iron is regarded as a scarce substrate in the natural environment, which is why our study focuses on the reduction of Fe(III) or the use of conductive minerals such as magnetite and hematite.

**I am also struggling to see how the authors incorporated advection or diffusion of dissolved Fe(II) into their models and interpretation. Depending on the rates of Fe(III) reduction, those would be a major controller of the extents of atom exchange (i.e., is Fe(II) exported quickly enough that no atom/electron exchange can occur?).**

We are actually not saying that there is NO atom/electron exchange. In fact, we even say in line 510 that it's very likely that these processes occur. In the paragraph ~line 585 we elaborate on this: "If part of the adsorbed (heavy) iron is then exchanged with the reactive Fe oxide surface (Crosby et al. 2007) and might subsequently even migrate deeper into the iron oxide crystal (Larese-Casanova et al., 2023), it could cause an alteration of Fe oxide isotope signatures towards positive values without reducing the mineral. It might also be speculated that adsorption and the related electron and atom exchange are more prevalent at depths that have a high Fe oxide ($Fe_{dith}$) content, but this interpretation remains very speculative, in particular because our model does not indicate adsorption to be a dominant Fe sink." What the model indicates is only that adsorption (and atom/electron exchange) is not the main process leading to the present pore-water and $\delta^{56}Fe$-profile.

**Despite these criticisms, I think the work is important because it contributes to what we know about patterns of Fe isotope compositions in different Fe pools – It's just hard to explain at this point. I think the authors have identified the major controller here: kinetics of Fe(III) reduction vs. kinetics of abiotic processes. I enjoyed reading this paper.**

**Specific comments:**

**Ln. 57. The authors do not include Fe(III) reduction by methanogens in their interpretation. The enzymology of that process is similarly understudied to that of fermenters.**

The reviewer mentions that we should include whether $Fe^{2+}$ release can also be linked to the reduction of $Fe^{3+}$ by methanogens that switch between methane generation and Fe reduction (e.g., Sivan et al. 2016, Eliani-Russak et al. 2023, Gupta et al. 2024 and references therein).

In the revised version, we will consider this aspect a bit more. Respiratory methanogenic iron reduction might be a possible explanation for deep Fe release in methanic sediments. However, as the process is respiratory, we assume (can't prove) that is would lead to similar Fe isotope fractionation as Fe reduction in shallow sediments. Furthermore, in order to do respiratory Fe(III) reduction, methanogens would need to oxidize $CH_4$ or an organic substrate (e.g., acetate, methyl compounds). Methane oxidation seems unlikely to support growth coupled to iron(III) reduction (see Chadwick et al., 2024). Which other electron donor is there for methanic zone methanogens to abandon their primary metabolism of $CO_2$ reduction with $H_2$ to methane? Gupta et al. (2024) summarize this issue by stating: "… even though we and others have shown that methanogens like *M. acetivorans* are metabolically active and can conserve energy by iron respiration […], robust growth that spans multiple generations is yet to be demonstrated i.e., it is still not known whether methanogens can couple iron reduction to growth in addition to energy conservation. Regardless, redox transformation of iron species by methanogens has substantial biogeochemical ramifications in and of itself to merit further investigation."

We are curiously following this very controversial discussion, but we have the feeling that this goes a bit beyond the scope of our study, which – in the end – is about stable Fe isotope signatures and does not contain microbiological data.

**ln. 345-351. avoid three sentence paragraphs. Also having trouble seeing how these observations are fitting into the broader story.**

We will revise this paragraph and give a better context concerning the Mn data.

**Ln. 620. this isn't completely true (benefitting both microbes). Unless the methanogen can use the reduced/conductive Fe phase as an electron donor, the Fe(III) just gets reduced and that's the end of it. This scenario is #1 on ln. 604.**

This comment refers to the following text: "The fermenting bacteria that transfer electrons to crystalline Fe oxides do not directly profit from Fe(III) reduction beyond the removal of thermodynamic limitations brought about by accumulation of fermentation intermediates. In other words: The fermenters use the conductive Fe oxides to transfer electrons and to be able to continue with the fermentation of particularly aromatic OM. The transfer of electrons via conductive Fe oxides speeds up the degradation of aromatic compounds and is beneficial to both partner microbes (e.g., Jiang et al., 2013; Kato et al., 2012; Zhuang et al., 2015). The transfer of electrons via conductive Fe oxides speeds up the degradation of aromatic compounds and **is beneficial to both partner microbes** (e.g., Jiang et al., 2013; Kato et al., 2012; Zhuang et al., 2015)."

The reviewer is right and this is exactly what we wanted to express. It's beneficial for both if the methanogen receives (part of) the electrons that are shuttled through the Fe oxide. In the revised version we will reformulate this to avoid confusion: "is **metabolically and mechanistically beneficial to both partner microbes** …".

**Ln. 622-624. I think Nathan Yee's group has done some work to address how fermenters reduce Fe(III).**

Thank you for this information. We checked and indeed found one paper by a member of his group that we might include into the discussion.

**Ln. 659-660. I agree with this, and think it's the strength of the paper.**

**Technical corrections:**

**Ln. 19. Please remove "unsurprisingly"**

**Ln. 386. Please change "conclusive" to "consistent"**

**Ln. 436. Please change "wit" to "with"**

Technical corrections will be done as suggested.

**Citation**: https://doi.org/10.5194/egusphere-2024-1942-RC1

**Aromokeye et al. (2021)** Crystalline iron oxides stimulate methanogenic benzoate degradation in marine sediment-derived enrichment cultures, ISME J, 15, 965–980, https://doi.org/10.1038/s41396-020-00824-7.

**Chadwick et al. (2024)** No evidence for methanotrophic growth of diverse marine methanogens. PNAS 121, 20, e2404143121, https://doi.org/10.1073/pnas.2404143121.

**Cruz Viggi et al. (2014)** Magnetite particles triggering a faster and more robust syntrophic pathway of methanogenic propionate degradation. Environ. Sci. Technol. 48, 7536–7543, https://doi.org/10.1021/es5016789.

**Dinh et al. (2004)** Iron corrosion by novel anaerobic microorganisms. Nature 427, 830-832, https://doi.org/10.1038/nature02321.

**Eliani-Russak et al. (2023)** The reduction of environmentally abundant iron oxides by the methanogen *Methanosarcina barkeri*. Front. Microbiol., 14, 1197299, https://doi.org/10.3389/fmicb.2023.1197299.

**Gupta et al. (2024)** MmcA is an electron conduit that facilitates both intracellular and extracellular electron transport in *Methanosarcina acetivorans*. Nat. Comm. 15, 3300, https://doi.org/10.1038/s41467-024-47564-2.

**Holmes et al. (2022)** Different outer membrane c-type cytochromes are involved in direct interspecies electron transfer to Geobacter or Methanosarcina species. mLife 1:3, 272-286, https://doi.org/10.1002/mlf2.12037.

**Kato et al. (2012)** Methanogenesis facilitated by electric syntrophy via (semi)conductive iron-oxide minerals. Environ. Microbiol. 14, 1646–1654, https://doi.org/10.1111/j.1462-2920.2011.02611.x.

**Sivan et al. (2016)** Methanogens rapidly transition from methane production to iron reduction. Geobiology, 14:2, 190-203, https://doi.org/10.1111/gbi.12172.

**Xu et al. (2019)** Enhancing direct interspecies electron transfer in syntrophic-methanogenic associations with (semi)conductive iron oxides: Effects and mechanisms. Sci. Total Environ. 695, 133876, https://doi.org/10.1016/j.scitotenv.2019.133876.